# Molecular mechanisms of exceptional lifespan increase of *Drosophila melanogaster* with different genotypes after combinations of pro-longevity interventions

Mikhail V. Shaposhnikov[1,6], Zulfiya G. Guvatova[2,6], Nadezhda V. Zemskaya[1], Liubov A. Koval[1], Eugenia V. Schegoleva[1], Anastasia A. Gorbunova[1], Denis A. Golubev[1], Natalya R. Pakshina[1], Natalia S. Ulyasheva[1], Ilya A. Solovev[1], Margarita A. Bobrovskikh[3], Nataly E. Gruntenko[3], Petr N. Menshanov [4,5], George S. Krasnov[2], Anna V. Kudryavtseva[2] & Alexey A. Moskalev [1,2✉]

Aging is one of the global challenges of our time. The search for new anti-aging interventions is also an issue of great actuality. We report on the success of *Drosophila melanogaster* lifespan extension under the combined influence of dietary restriction, co-administration of berberine, fucoxanthin, and rapamycin, photodeprivation, and low-temperature conditions up to 185 days in $w^{1118}$ strain and up to 213 days in long-lived *E(z)/w* mutants. The trade-off was found between longevity and locomotion. The transcriptome analysis showed an impact of epigenetic alterations, lipid metabolism, cellular respiration, nutrient sensing, immune response, and autophagy in the registered effect.

[1] Laboratory of Geroprotective and Radioprotective Technologies, Institute of Biology, Komi Science Centre, Ural Branch, Russian Academy of Sciences, Syktyvkar 167982, Russian Federation. [2] Center for Precision Genome Editing and Genetic Technologies for Biomedicine, Engelhardt Institute of Molecular Biology, Russian Academy of Sciences, Moscow 119991, Russian Federation. [3] Laboratory of Stress Genetics, Institute of Cytology and Genetics, Siberian Branch of Russian Academy of Sciences, Lavrentiev avenue 10, Novosibirsk 630090, Russian Federation. [4] Physiology Department, Novosibirsk State University, Pirogova 1, Novosibirsk 630090, Russian Federation. [5] Laser Systems Department, Novosibirsk State Technical University, Karl Marx avenue 20, Novosibirsk 630073, Russian Federation. [6] These authors contributed equally: Mikhail V. Shaposhnikov, Zulfiya G. Guvatova. ✉email: amoskalev@ib.komisc.ru

Aging is a complex process characterized by the decline of biological functions and the gradual decrease in resistance to multiple stresses, which leads to age-related pathologies and eventually causes death[1,2]. Over the past decades, a growing body of researches in biogerontology has shown that various interventions, such as genetic manipulations[3,4], light regimens[5], reduction in core body temperature[6], dietary modulations[7,8], and pharmacological agents targeting the age-related molecular and cellular processes (anti-aging drugs or geroprotectors)[9–12] could extend healthspan and lifespan in various model organisms.

Caloric (CR) and dietary restrictions (DR) are the most studied metabolic interventions for which a role in the lifespan extension of different model organisms from yeast to non-human primates are shown[7,8]. Notable that although low body temperature is usually attributed to the reduction of calorie intake there are studies demonstrating that reduction in core body temperature could influence longevity independently of CR both in poikilotherms and homeotherms[6,13]. The negative correlation between temperature and longevity has been discovered in the nematodes, rotifers, fruit flies, and in several killifish species[14–16]. Moreover, several long-lived mutant mice have shown a decrease in core body temperature, which supports the idea of maintaining the positive effect in mammals[13,17].

The usage of pharmacological approaches also emerges as a promising anti-aging strategy. According to the recent experimental data, a combination of several anti-aging pharmacological treatments has greater effects on health and lifespan compared to single treatments[18–22]. Indeed, accumulated evidence makes it clear that the most effective interventions come down to the regulation of only a few cellular processes, in particular nutrient signaling, mitochondrial efficiency, proteostasis, and autophagy. Multiple treatments that target these aging-related signaling pathways appear to be a potential approach for delaying aging processes and increasing lifespan[23].

In this study, we investigated whether the combined application of several interventions with potential anti-aging action causes a cumulative effect on lifespan extension. As for anti-aging drugs, we used rapamycin, the well-known mTOR signaling inhibitor, and two plant-derived compounds, particularly, alkaloid berberine and carotenoid fucoxanthin, whose geroprotective properties have been studied on different biological models[24–27].

We studied the effects of DR and co-administration of berberine, fucoxanthin, and rapamycin in constant darkness and low-temperature conditions using the *D. melanogaster* model. In addition, to address whether the long-lived strain demonstrates an enhanced geroprotective effect of the interventions' combinations, we studied the long-lived *Enhancer of zeste (E(z))* mutant flies. E(z) is the catalytic subunit of Polycomb Repressive Complex 2 (PRC2) with H3K27 trimethylation activity. Heterozygous loss-of-function mutation in *E(z)* has been shown to disrupt Polycomb silencing and lead to increased resistance to various types of stress and, as a result, to increased lifespan[28,29].

Thus, in the current study, we observed significant changes in lifespan, locomotor activity, and stress resistance in flies of both genotypes in response to combinations of anti-aging interventions and investigated the underlying mechanisms of these effects using analysis of the whole-genome transcriptome, retrotransposons activity, and total lipid content.

## Results

**Lifespan**. The effects of different experimental conditions (maintaining in the dark (DD), low ambient temperature (18 °C), exposure to geroprotectors (3G), dietary restriction (DR)) with potential geroprotective activities and its possible combinations

on the lifespan of the *w/w* control line and long-lived mutants *E(z)/w* were analyzed.

The strongest increase in longevity was observed in *w/w* and *E(z)/w* flies which were kept in a combination of all studied factors (Figs. 1a–d and 2c, d; Supplementary Table 1; Supplementary Data 1). More specifically, the DR, 3G, 18 °C, DD combination substantially ($p < 0.001$) increased median lifespan (*w/w* males: 153 days, by 164%; *E(z)/w* males: 171 days, by 122%; *w/w* females: 149 days, by 126%; *E(z)/w* females: 165 days, by 88%), maximum lifespan (*w/w* males: 184 days, by 104%; *E(z)/w* males: 213 days, by 124%; *w/w* females: 185 days, by 115%; *E(z)/w* females: 200 days, by 90%), and shifted survival curves rightward ($p < 0.001$) compared to controls.

It should be noted that among the group of experimental variants that were characterized by the most significant increase in maximum lifespan (*w/w* males: by 68–102%; *w/w* females: by 67–95%; *E(z)/w* males: by 73–107%; *E(z)/w* females: by 56–87%), all were kept in low ambient temperature conditions.

Contrary to our hypothesis, some used factors when used singly and in combination, had a statistically significant negative effect on lifespan. Specifically, DR decreased the age of 90% mortality and maximum lifespan of the *w/w* control line (males: by 9% and 4%; females: by 6% and 3%, respectively), and all experimental variants without low ambient temperature decreased survival in *E(z)/w* flies (Figs. 1a–d and 2c, d; Supplementary Table 1; Supplementary Data 1). Treatment of *E(z)/w* flies with 3G caused the greatest decrease in median lifespan (males: by 36%; females: by 30%) and the age of 90% mortality (males: by 26%; females: by 15%).

At the same time, some interventions did not induce a statistically significant effect on median lifespan or age of 90% mortality (*w/w* males: 3G; 3G, DD; DR, 3G, DD; *w/w* females: 3G, DD; *E(z)/w* males: DR, 3G, DD; *E(z)/w* females: DR, 3G, DD).

The positive, negative, or neutral effect of dietary and pharmacological interventions may be associated with the general nature of hormetic action of dietary proteins[30] or anti-aging drug[31] concentrations in the nutrition medium, and maybe substantially modified by sex, genotype, and other environmental factors[32,33].

When the contribution of individual conditions in all used combinations was addressed, Cox regression analysis showed that 18 °C is associated with a strongly reduced risk of death in both *w/w* and *E(z)/w* flies (Figs. 1e, f and 2e, f; Supplementary Table 2; Supplementary Data 1). The hazard ratio (HR) of death associated with the 18 °C in *w/w* males and females was 0.282 and 0.090, respectively, corresponding to 3.5 and 11-fold reduced mortality risk, respectively ($p < 0.001$). The HR of 0.036 and 0.094 for 18 °C in *E(z)/w* male and female flies, respectively, correspond to 27.4- and 10.6-fold reduced mortality risk, respectively ($p < 0.001$).

While the contribution to the lifespan effects of DD and DR were non-significant ($p > 0.05$) in *w/w* males, the HR of 0.847 ($p < 0.001$) and 0.602 ($p < 0.001$) for DD and DR in *w/w* females demonstrated decreased risks of death, up to a 1.2- and 1.7-fold respectively (Figs. 1e, f and 2e, f; Supplementary Table 2; Supplementary Data 1). However, in *E(z)/w* flies the HR for DD (males: 0.698; females: 0.756) and DR (males: 0.816; females: 0.781) demonstrated significantly ($p < 0.001$) decreased (up to 1.4- and 1.3-fold, respectively) risks of death. Cox regression analysis indicated significantly reduced risks of death for 3G in *w/w* males (HR of 0.891, $p < 0.05$) and females (HR of 0.657, $p < 0.001$). At the same time, 3G is associated with increased risks of death in *E(z)/w* males (HR of 1.483, $p < 0.001$) and females (HR of 1.217, $p < 0.001$). The observed sex- and the genotype-dependent relationship between longevity and other studied factors, including lighting[34], diet[30,32], and pro-

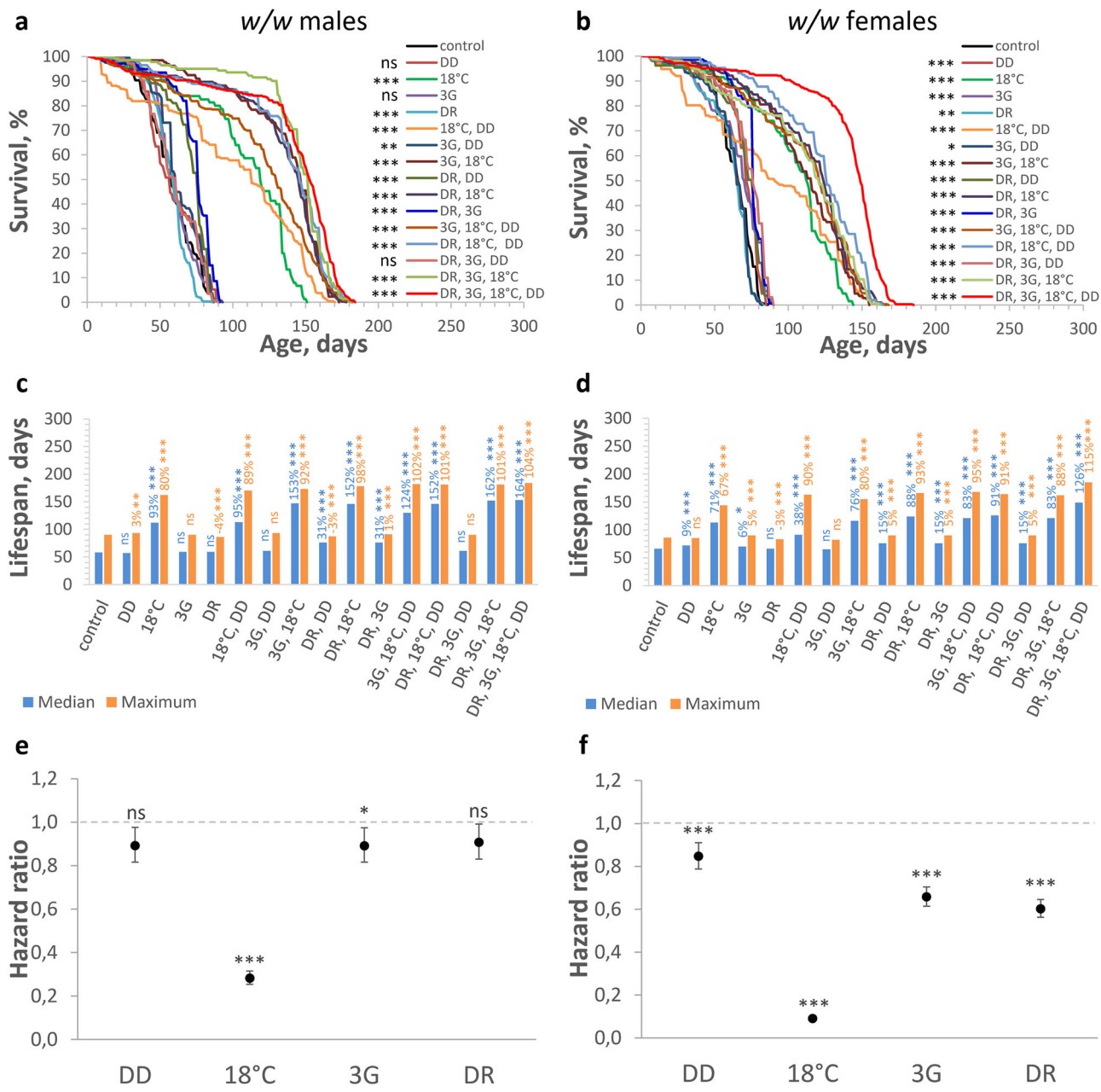

**Fig. 1 Effects of different factor combinations (maintaining in the dark (DD), low ambient temperature (18 °C), a combination of rapamycin, berberine, and fucoxanthin (3G), dietary restriction (DR)) on lifespan parameters in *w/w* flies. a**, **c**, **e** Males and (**b**, **d**, **f**) females. **a**, **b** Survival curves. **c**, **d** Median and maximum lifespan. **e**, **f** Cox proportional hazards regression model with hazard ratios and 95% confidence intervals. Dashed lines indicate a hazard ratio of 1, which corresponds to control conditions (12 h light: 12 h dark, 25 °C, no substances added, normal diet). Asterisks (*) indicate the level of statistical significance of differences (*$p < 0.05$; **$p < 0.01$; ***$p < 0.001$, log-rank test (**a**, **b**), Fisher's exact test (**c**, **d**), Cox proportional hazards regression (**e**, **f**)); statistical significance of differences in maximum lifespan is based on the $p$ values for the age of 90% mortality; ns – not significant; $n = 300$ flies. Bonferroni correction was used in all multiple comparisons. All source data underlying the graphs and charts are presented in the Supplementary Data 1.

longevity chemicals[33] are consistent with previously published experimental data.

The significant lifespan-extending effects of low-temperature conditions are also consistent with the many studies conducted on invertebrate and vertebrate models[13–17]. Previously published studies have shown that the longest maximum lifespan of *D. melanogaster* (wild-type *Oregon-R* males) at 18 °C varies within the range of 163–180 days[35,36]. The reported values of maximum lifespan correspond to the effects of 18 °C on the maximum lifespan of control *w/w* flies (162 days in males and 144 days in females) and long-lived *E(z)/w* flies (194 days in males and 174 days in females), but they are substantially lower

than maximum lifespan at DR, 3G, 18 °C, DD combination (up to 185 and 213 days in *w/w* and *E(z)/w* males, respectively).

Despite the mechanisms underlying the effect of low temperature remaining largely unknown, it is assumed that in addition to the thermodynamic process, changes in the expression of genes affecting the rate of aging may contribute to life extension[16,37]. The dietary restriction[38], drug combination[19,20], and light regime[39] have also been shown to influence expression of longevity-related genes.

Taking into account that the contribution of temperature factor significantly exceeds the effects of other interventions, Cox regression analysis was performed separately for control (25 °C)

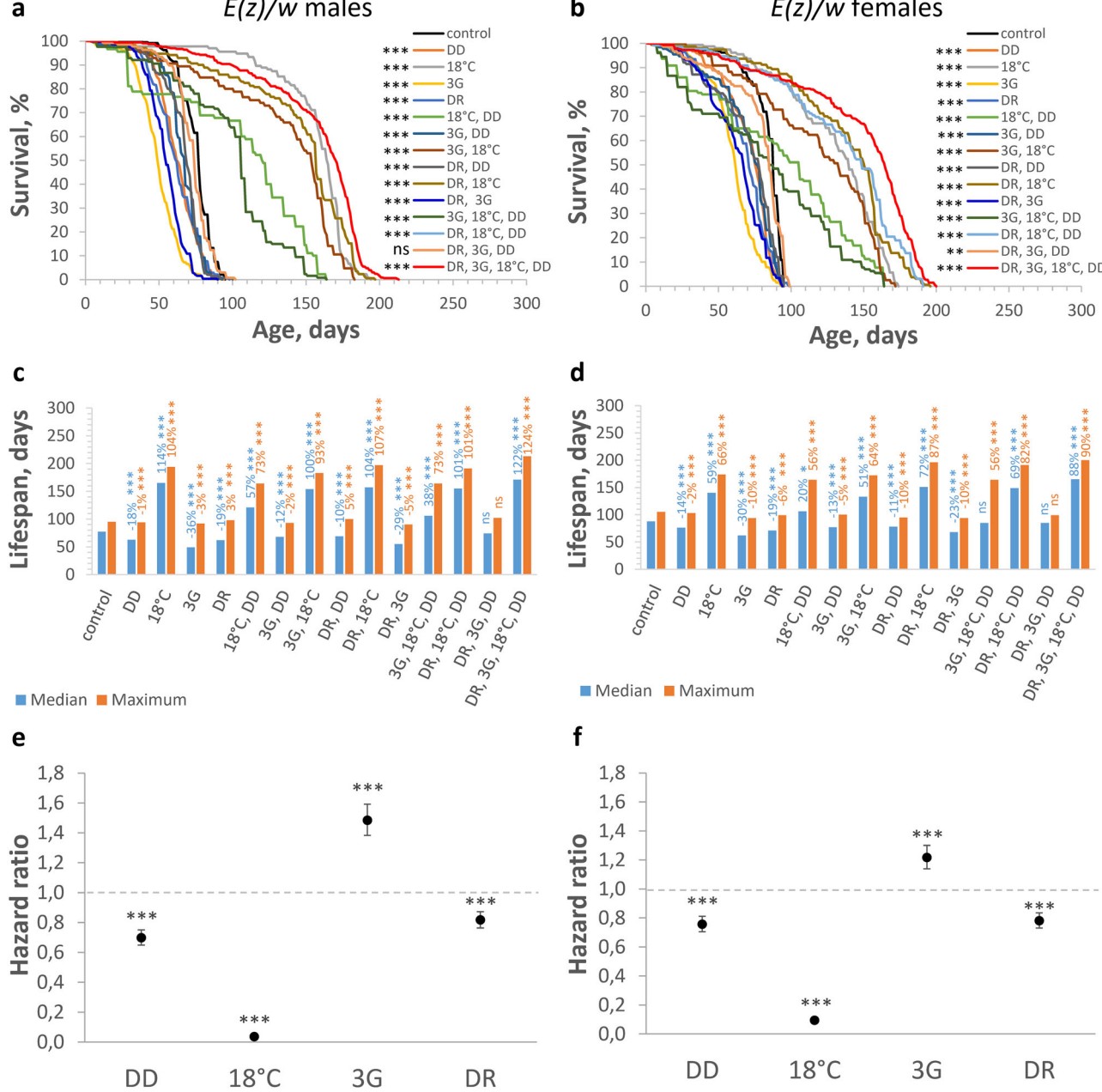

**Fig. 2 Effects of different factor combinations (maintaining in the dark (DD), low ambient temperature (18 °C), a combination of rapamycin, berberine, and fucoxanthin (3G), dietary restriction (DR)) on lifespan parameters in *E(z)/w* flies. a**, **c**, **e** Males and (**b**, **d**, **f**) females. **a**, **b** Survival curves. **c**, **d** Median and maximum lifespan. **e f**, Cox proportional hazards regression model with hazard ratios and 95% confidence intervals. Dashed lines indicate a hazard ratio of 1, which corresponds to control conditions (12 h light: 12 h dark, 25 °C, no substances added, normal diet). Asterisks (*) indicate the level of statistical significance of differences (*$p < 0.05$; **$p < 0.01$; ***$p < 0.001$, log-rank test (**a**, **b**), Fisher's exact test (**c**, **d**), Cox proportional hazards regression (**e**, **f**)); statistical significance of differences in maximum lifespan is based on the $p$ values for the age of 90% mortality; ns not significant; $n = 300$ flies. Bonferroni correction was used in all multiple comparisons. All source data underlying the graphs and charts are presented in the Supplementary Data 1.

and experimental (18 °C) temperature variants to reveal the condition-dependent lifespan effects of DD, DR, 3G, sex, and genotype (Supplementary Fig. 1; Supplementary Table 3). The risks of death were up to 1.1–1.4 and 1.2–1.3 lower for DR and *E(z)/w* genotypes at 25 °C and 18 °C, respectively, as compared to the reference conditions, demonstrating the contributions of diet and genotype to the lifespan effects were not temperature-dependent. However, DD was significantly associated with increased lifespan (HR of 0.765, $p < 0.001$) at 25 °C only, while the contribution of 3G and sex to the lifespan effects were

opposite at different temperatures. The 3G was significantly associated with increased (HR of 1.199, $p < 0.001$) or decreased (HR of 0.871, $p < 0.001$) risks of death at 25 °C and 18 °C, respectively (Supplementary Fig. 1; Supplementary Table 3). Contrary, female sex was linked with decreased (HR of 0.624, $p < 0.001$) or increased (HR of 1.164, $p < 0.001$) risks of death at 25 °C and 18 °C, respectively.

Together, these results demonstrate that while the separate interventions can increase the general risk of death and decrease longevity depending on the sex, and genotype, the combined

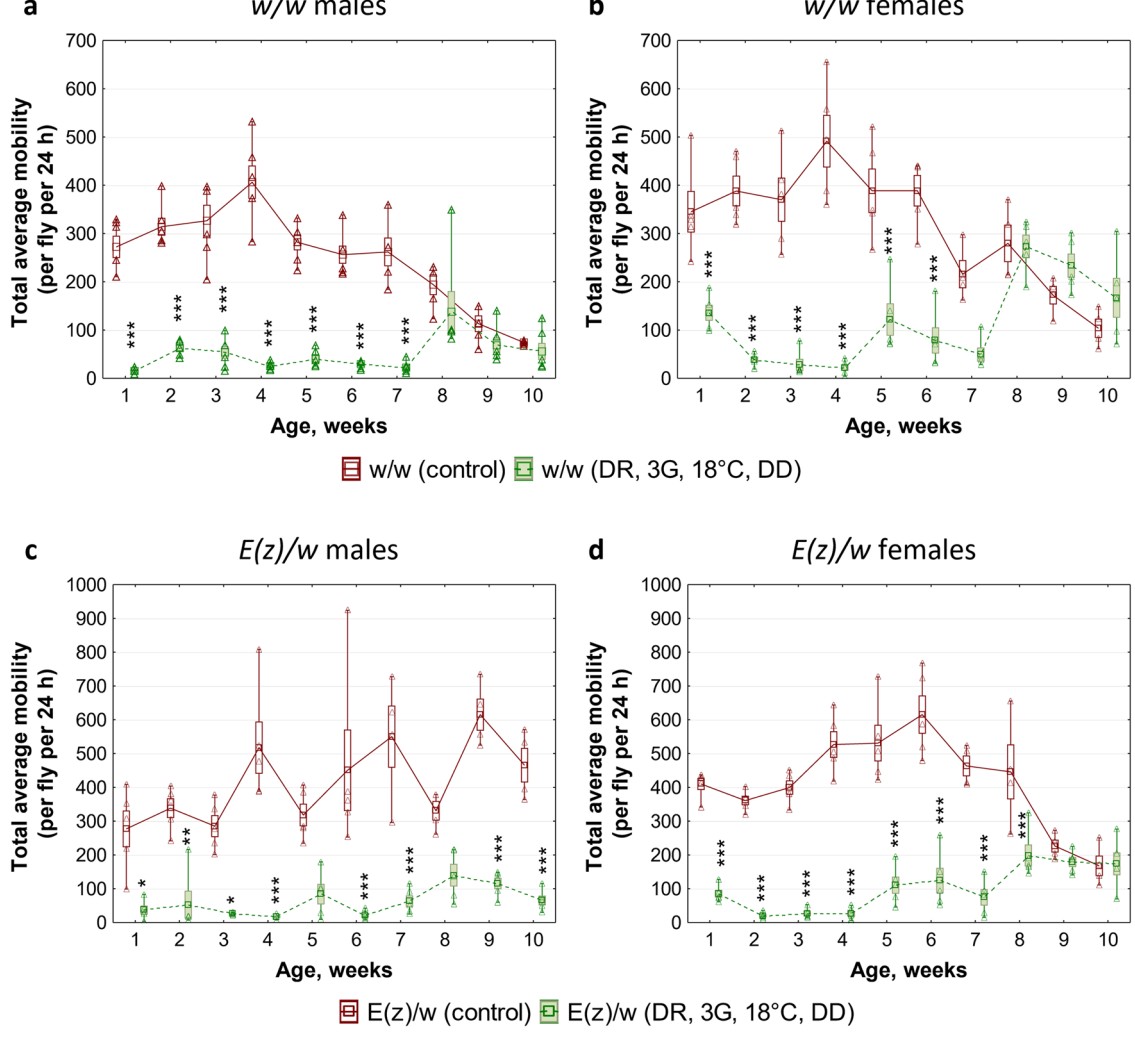

**Fig. 3 Effects of a combination of dietary restriction (DR), geroprotectors (3G), low ambient temperature (18 °C), and maintenance in the dark (DD) on age-dependent dynamics of total daily locomotor activity. a**, **b** Control line *w/w*. **c**, **d** Long-lived line *E(z)/w*. **a**, **c** Males. **b**, **d** Females. The plots show individual data points (triangles), means (squares), standard errors (boxes), minimum and maximum values (whiskers). Asterisks (*) indicate the level of statistical significance of differences (*$p < 0.05$; **$p < 0.01$; ***$p < 0.001$, two-way ANOVA followed by post-hoc Tukey's HSD tests); $n = 50$ flies. All source data underlying the graphs are presented in the Supplementary Data 1.

action of all studied interventions significantly reduces the general risk of death and significantly enhance the subject's longevity. The low-temperature factor makes the greatest contribution to longevity.

**Locomotor activity**. The effects of combined intervention (DR, 3G, 18 °C, DD) on age-dependent dynamics of locomotor activity of both *w/w* and *E(z)/w* flies were analyzed as the indicators of healthspan[40]. Two-way analysis of variance (ANOVA) followed by post-hoc Tukey's HSD tests revealed a statistically significant ($p < 0.05$) decrease in locomotor activity of flies that were kept under the combination of experimental conditions from the age of 1 to 6–7 weeks (Fig. 3a–d; Supplementary Table 4; Supplementary Data 1). In the control groups (*w/w* males and females and *E(z)/w* females) a decrease in locomotor activity to the level of activity of experimental flies (Fig. 3a, b, d; Supplementary Data 1) at the age of 8–10 weeks was found, which indicates a more pronounced aging process in the control groups. At the same time, a significantly delayed age-related decline of loco-motor performance was found in *w/w* and *E(z)/w* flies from the experimental groups. It is noteworthy that the long-lived *E(z)/w*

males from the control group have also demonstrated post-ponement of age-related decline in locomotor activity (Fig. 3c; Supplementary Data 1).

To address which of the used interventions make the greatest contribution to the level of locomotor activity, we analyzed the motility of flies under various combinations of experimental factors (Supplementary Fig. 2). Spearman's correlation analysis (Supplementary Table 5) revealed moderate (−0.476 and −0.610) but very highly significant ($p < 1 \times 10^{-17}$) associations between the level of locomotor activity and two conditions (18 °C and DD, respectively). Evaluation of the contribution of low ambient temperature (18 °C) and absence of light (DD) to the locomotor activity was performed using Bayesian ANOVA, taking into account differences in sex, age, and genotype. The model comparison revealed the most probable contribution of factor combination: GT + DD + 18 °C (Supplementary Table 6).

The analysis of covariance (ANCOVA)[41] was used to estimate the relative strength of the contribution of DD and 18 °C versus sex, age, and genotype (Supplementary Table 7). Partial eta-squared values suggest that the main effect on locomotor activity is associated with maintaining in the dark (DD: $\eta^2 = 0.188$, $p < 0.001$) and, to a lesser extent, genotype (GT: $\eta^2 = 0.007$, $p < 0.001$) and low

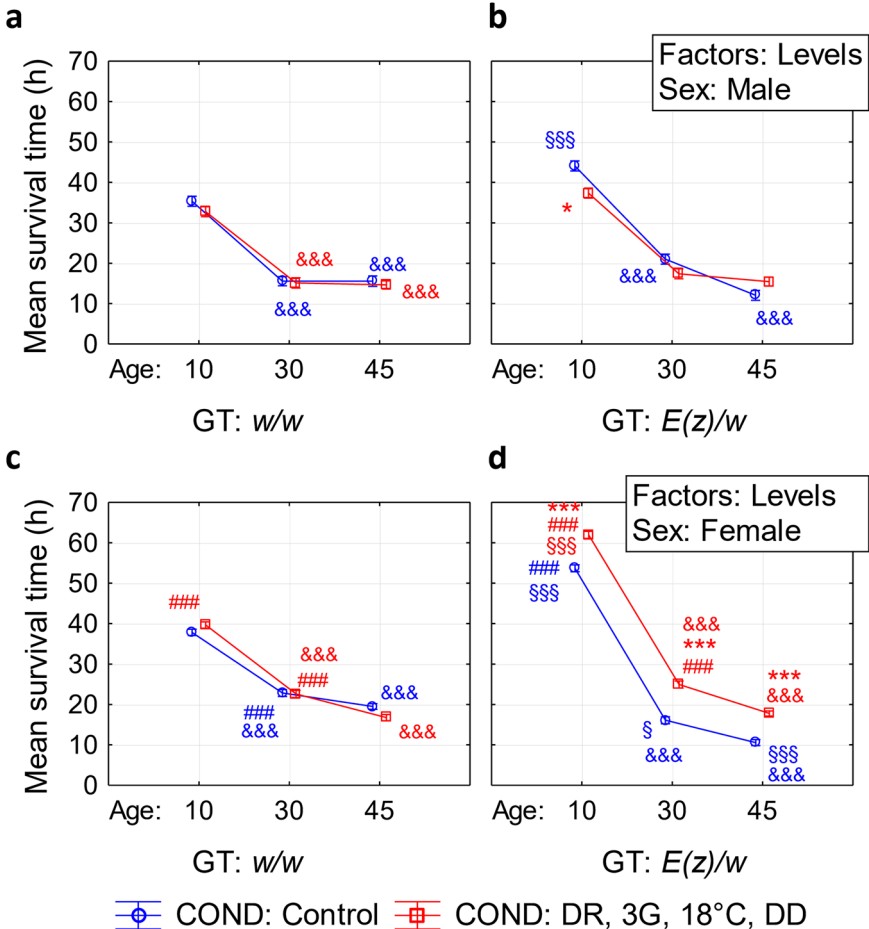

**Fig. 4 Four-way ANOVA analysis of differences in resistance to oxidative stress depending on sex (SEX), age (AGE), genotype (GT), and combination of interventions (COND), including dietary restriction (DR), geroprotectors (3G), low ambient temperature (18 °C), and maintenance in the dark (DD).** a *w/w* males, b *E(z)/w* males, c *w/w* females, d *E(z)/w* females. $^{###}p < 0.001$—differences between males and females; $^{§}p < 0.05$, $^{§§§}p < 0.001$—differences between *w/w* and *E(z)/w* genotypes; $^{*}p < 0.05$; $^{**}p < 0.01$; $^{***}p < 0.001$—differences between control and experimental maintaining conditions; $^{&&&}p < 0.001$—differences between ages. The level of statistical significance of differences was revealed by four-way ANOVA with poct-hoc Bonferroni tests. The error bars show standard error; $n = 96$ flies. All source data underlying the graphs are presented in the Supplementary Data 1.

ambient temperature (18 °C: $\eta^2 = 0.006$, $p < 0.001$). At the same time, effect size of sex (SEX: $\eta^2 = 0.003$, $p < 0.05$) and age (AGE: $\eta^2 = 0.002$, $p < 0.05$) were minimal.

These results indicate that the combination of interventions significantly prolongs the healthspan of flies. However, an increase in lifespan is associated with negative side effects of low-temperature conditions and constant darkness on locomotor activity.

**Stress resistance**. To assess whether the observed lifespan-extending effect of DR, 3G, 18 °C, DD combination was associated with increased stress tolerance, resistance to oxidative stress was analyzed. Partial eta squared values which were obtained through four-way ANOVA analysis (Supplementary Table 8) demonstrated that the largest effects on stress resistance ($p < 0.001$) are associated with age (AGE: $\eta^2 = 0.559$), sex (SEX: $\eta^2 = 0.065$), genotype (GT: $\eta^2 = 0.028$), and interaction between the effects of these factors (GT×AGE: $\eta^2 = 0.088$; AGE×SEX: $\eta^2 = 0.032$; GT×AGE×SEX: $\eta^2 = 0.032$).

Bayesian analysis of four-way ANOVA model revealed the following combination of factors as the most probable model: SEX + GT + COND + AGE + SEX×GT + SEX×COND + GT×COND + SEX×AGE + GT×AGE + SEX×GT×COND + SEX×GT×AGE (Supplementary Table 9).

The largest effects of sex and age (SEX, SEX×AGE at AGE) demonstrate that females *w/w* were more tolerant to paraquat than males, and the resistance of both sexes decreased with age (Fig. 4; Supplementary Fig. 3; Supplementary Table 9 and 10; Supplementary Data 1). Identified interaction of sex, genotype, and age (SEX×GT×AGE) suggests that with age, the *E(z)* mutation reduces resistance to paraquat in both females and males (GT×AGE at GT), while in *E(z)/w* females stress-resistance decreases at an earlier age than in males (SEX×GT×AGE at SEX×GT). In turn, the interaction of sex, genotype, and conditions (SEX×GT×COND) shows that the combination of experimental factors restores (and even increases at an early age) the level of resistance of flies *E(z)/w* of both sexes to paraquat (SEX×GT× COND at COND, SEX×COND, GT×COND).

Despite a significant lifespan increase in all experimental variants, we did not observe increased resistance to oxidative stress in most variants and ages except *E(z)/w* females. Thus the lifespan-extending effect of studied anti-aging interventions combination is not strongly associated with stress tolerance.

**Total lipid contents**. Fat metabolism has a crucial role in stress resistance[42] and lifespan[43]. To investigate how fat metabolism in adult *w/w* and *E(z)/w* flies is influenced by the combination of conditions, we assessed the total lipid contents in flies. Significant

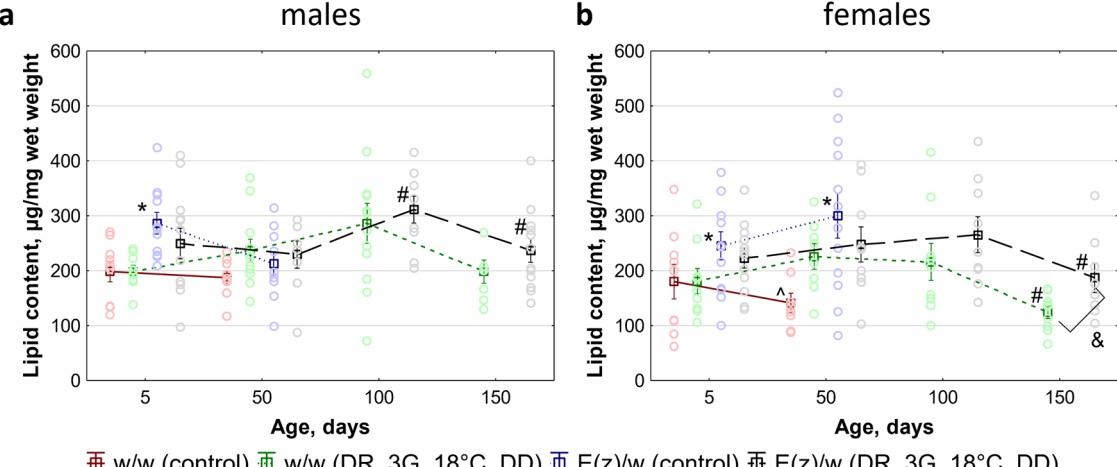

**Fig. 5 Effect of combination of anti-aging interventions (dietary restriction (DR); co-administration of berberine, fucoxanthin, and rapamycin (3G); low ambient temperature (18 °C); living in constant darkness (DD) conditions) on the level of total lipids in *w/w* and *E(z)/w* flies. a** Males. **b** Females. Control conditions: normal diet, 25 °C, and 12 h light: 12 h dark cycle. Each data point represents a mean of 8–12 biological replicates ± SEM. Plots show individual data points (circles), means (squares), and standard errors (vertical lines). *$p < 0.05$ compared with age-matched *w/w* flies, #$p < 0.05$ compared with the flies of the previous age point, ^$p < 0.05$ compared with age-matched *w/w* flies on the same conditions, &$p < 0.05$ compared with *w/w* flies on the same conditions. All source data underlying the graphs are presented in the Supplementary Data 1.

differences in the levels of total lipid contents were found between males and females (Supplementary Fig. 4). Further analysis of the effects of experimental conditions on total lipid levels was performed for each sex separately (Fig. 5; Supplementary Data 1).

The total lipid levels are found to be principally dependent on genotype and age both in males (Supplementary Table 11) and females (Supplementary Table 12; GT: $F_{(1.68)} = 6.23$, $p = 0.016$; AGE: $F_{(3.68)} = 4.76$, $p = 0.0046$). The possible effects of the interventions' combination on the total lipid levels were less evident and most likely depended on other principal factors (Supplementary Tables 11 and 12).

To ensure between-sex comparability (Supplementary Fig. 5), we normalized absolute lipid levels to body weight and obtained relative lipid levels (Supplementary Fig. 4). The differences between the mean values of relative total lipid levels in both sexes did not exceed 20%.

Age-related changes in the total lipid levels were determined by genotype ($F_{(1.142)} = 6.29$, $p = 0.014$), age ($F_{(3.142)} = 6.30$, $p = 0.00049$), and sex ($F_{(1.142)} = 9.13$, $p = 0.003$) independently (Supplementary Table 13). Any of these factors influenced the total lipid content as an additive one, with minimal interactions between them. The age-related decrease in the total lipid content in *w/w* females was delayed, but could not be stopped by interventions' combinations. Regardless of the total lipid content loss, these females remained alive up till the 150th day of life. In other flies that received a combination of interventions, the age-specific loss of lipids was less evident.

Thus, the main factors affecting total lipid contents are sex, age, and genotype. The contributions of these factors are independent and additive, without significant non-additive effects. The effect of conditions combination on lipid concentration is insignificant and the lifespan-extending effect seems to be not associated with total lipid contents.

**Retrotransposon activity.** To address whether transposition activity is affected by flies' age and combination of all used interventions (DR, 3G, 18 °C, DD) and we analyzed the expression of 9 retrotransposon families (*Het-A1, R1, Rt1α, 1731, 412, blood, opus, roo,* and *LINE-1*). Taking into account that the combinations of used factors and *E(z)* mutation, may target

different tissues[44–46] including the nervous system, intestine, and fat body for which age-dependent activation of retrotransposons was shown[47–51] the levels of transpositions were evaluated in fly whole bodies.

Four-way Bayesian analysis of variance with covariate (ANCOVA) revealed a significant specificity of the levels of expression of retrotransposons depending on sex (Supplementary Table 14), and the subsequent three-way Bayesian ANCOVA identified general patterns of retrotranspositions for each sex (Supplementary Table 15).

The most probable model in males is AGE + GT + COND + ANCOVA + AGE×GT + AGE×COND + GT×COND + AGE×GT×COND (Supplementary Table 15). The levels of retrotransposon activity in 10-day-old male flies maintained in the control conditions were not affected by *E(z)/w* genotype (Supplementary Fig. 6). However, under experimental conditions, a significant decrease in the levels of retrotransposon activity was observed in 10-day-old *E(z)/w* males but not in *w/w* males compared to the *E(z)/w* and *w/w* males, respectively, maintained in the control conditions (AGE×GT×COND at GT×COND, AGE×COND and COND).

The levels of activity of most retrotransposons in *w/w* and *E(z)/w* males maintained in the control conditions at the age of 30 days old were comparable, except for the higher level of activity of *1731* and *blood* in *E(z)/w* males (AGE×GT at GT and AGE). The experimental conditions had no significant effects on the activity levels of all studied transposons in 30-day-old male flies.

The most probable model in females is AGE + GT + ANCOVA + AGE×GT (Supplementary Table 15). The levels of activity of each of the retrotransposons demonstrated a weak relationship with the genotype (Supplementary Fig. 7), except for more active retrotransposons *1731* and *blood* in 30-day-old females *E(z)/w* (AGE×GT at AGE and GT). Similar but not statistically significant ($p > 0.05$) changes were observed for the *HetA* transposon. Under the combination of AGE and GT experimental factors, the levels of *1731* and *blood* activity were partly decreased in *E(z)/w* females at 30 days of age but did not reach the same levels as in experimental 30-day-old *w/w* females.

Thus, the most pronounced effect of experimental factors combination on the activity of retroelements was observed in

10-day-old *E(z)/w* males. These results suggest that no clear relationship was found between the activity of retrotransposons and lifespan effects in used experimental conditions.

**Transcriptome analysis.** Next, we performed RNA-Seq to analyze the effects of DR, 3G, 18 °C, DD combination on gene expression profiles of *w/w* and *E(z)/w* flies. To assess the transcriptomic changes we used treated female and male flies of both genotypes at the age of 5, 50, 100, and 150 days old and untreated control flies at the age of 5, 50 days old. The untreated(control) flies did not survive to 100 and 150 days. All analyzed variants are presented in Supplementary Table 16. According to differential expression analysis the combination of interventions caused more than two-fold changes in expression ($|LogFC| \geq 1$, $LogCPM > 1$) of 31 genes in 50-day-old *w/w* females ($p < 0.05$, no genes passed $FDR < 0.05$) and 772 genes in 50-day-old *w/w* males (among them 745 genes with $FDR < 0.05$). At the same time, the changes in long-lived *E(z)/w* flies of the same age were more dramatic: 556 (523 with $FDR < 0.05$, $|LogFC| \geq 1$, $LogCPM > 1$) differentially expressed genes (DEGs) in females and 1396 DEGs (1372 with $FDR < 0.05$, $|LogFC| \geq 1$, $LogCPM > 1$) in males were detected. In 5-day-old flies, the number of DEGs was approximately the same in both genotypes: 388 (377 with $FDR < 0.05$, $|LogFC| \geq 1$, $LogCPM > 1$) DEGs in *w/w* females, 579 (574 with $FDR < 0.05$, $|LogFC| \geq 1$, $LogCPM > 1$) DEGs in *w/w* males, 502 (487 with $FDR < 0.05$, $|LogFC| \geq 1$, $LogCPM > 1$) DEGs in *E(z)/w* females, 485 (479 with $FDR < 0.05$) DEGs in *E(z)/w* males. Comparing treated 100- and 150-day-old flies of different genotypes (*E(z)/w* versus *w/w*), we detected a strong difference ($FC > 8$) in the expression of a number of genes, for most of which we have previously observed the association with the *E(z)* mutation[28]. Complete lists of DEGs are presented in Supplementary Data 2.

To determine what cellular processes are affected by selected interventions, we performed a KEGG pathway enrichment analysis using DEGs between treated and untreated flies of both genotypes. In males of both genotypes and *E(z)/w* females the statistically significant enrichment ($FDR < 0.05$) with upregulated genes was observed for "Oxidative phosphorylation" pathway, suggesting a significant effect of studied interventions on energy metabolism. In males of both genotypes, we found a change in the "Hippo signaling pathway" involved in several key mechanisms associated with the aging processes, such as sirtuin pathways, autophagy, and oxidative stress response[52]. It is also worth noting the effect on lipid metabolism: alterations in "Sphingolipid metabolism", "Fatty acid metabolism", and "Fatty acid degradation" pathways have been detected (Fig. 6).

Transcriptomic profiling of 50-days-old (Supplementary Fig. 8) and 5-days-old flies revealed a similar set of biological processes and cellular pathways that were most likely affected. However, according to the analysis of 5-days-old flies, there were no pathways associated with lipid metabolism among statistically significantly enriched pathways, which is consistent with previously described lipid analysis results.

Next, we focused on identifying pathways enriched with DEGs that change their expression during flies' aging (Fig. 7). The most pronounced alterations are observed in glutathione, drugs, and xenobiotic metabolism. Nucleotide and fatty acid metabolism are also affected during aging. Our results suggest the age-dependent decrease in expression of genes related to "Oxidative phosphorylation", and "Citrate cycle (TCA cycle)" pathways in *w/w* flies and, to a lesser degree, in *E(z)* mutants. In the males of both genotypes, we found the age-dependent up-regulation of "Ribosome" pathway genes and down-regulation of "Peroxisome" pathway genes, while in the females such changes were not detected.

Next, we compared our derived lists of DEGs with *Drosophila* genes presented in the GenAge database, the database of genes related to longevity and/or aging in model organisms and humans[53]. During aging, the combination of all studied factors influenced the change in expression of 95 genes in *E(z)/w* males, 54 genes in *E(z)/w* females, 91 genes in *w/w* males, 61 genes in *w/w* females, for which an association with the lifespan of *Drosophila* was previously revealed according to GenAge. The direction (increase or decrease in expression during aging) and LogFC values of detected DEGs, except for the *Tor* gene, had a similar pattern in *E(z)/w* and *w/w* males. The expression of *Tor* declines with age in both groups, however, the magnitude of change is more pronounced in the *E(z)* mutants (LogFC was −1.9 versus −0.8 in *w/w* males). As a participant of the mTOR pathway, Tor plays one of the key roles in maintaining energy homeostasis and its modulation influences longevity and aging[54]. In females kept under experimental conditions, we observed a change in the expression of the *mys* (*myospheroid*) gene encoding a β subunit of the integrin dimer. It was shown that loss-of-function mutations in *mys* protect flies from age-related defects in locomotor behavior[55]. Notably, *E(z)* females had a stronger decrease in expression (LogFC = −0.97) than wild-type females (LogFC = −0.41).

In *w/w* females, we also found an age-related increase in the expression of the *bam (bag of marbles)* gene involved in gametogenesis in flies. Previously, Flatt, et al[56]. have linked the lifespan extension of *Drosophila bam* overexpressing mutants with the loss of germ cells. According to our results, there was no statistically significant change in *bam* expression in *E(z)* females. Indeed, in previous work, we have shown that long-lived *E(z)/w* flies had higher fecundity compared to the control *w/w* flies[28].

The expression level of age-related genes such as *Pink1*, *CG3776*, *Esc*, *park*, *CG14207*, *Surf1* was altered in experimental *E(z)* mutant males, while no statistically significant changes were found in the experimental *w/w* flies. The *Pink1* showed a pattern of mRNA expression that, as previously shown, positively correlates with an increase in the lifespan of model organisms.

The next step in the transcriptome analysis was a comparison of data from the GenAge database with the results of comparisons (DEG) of experimental 50-day-old flies (including both of *E(z)/w* and *w/w* flies) with the corresponding control groups. We revealed a number of genes associated with the FoxO signaling pathway (*foxo*, *bsk*, *Akt1*, *SNF4Agamma*, *Ilp*, *Cat*) and autophagy (*Atg1*, *Atg2*, *Atg7*, *Atg8A*, *Pi3K92E*). Interestingly, the expression level of the autophagy-related genes (*Atg*) was reduced. The expression of aging-associated genes *CG3776*, *p53*, *Sirt6*, *rut*, *E(z)*, *foxo*, *Thor*, *Mad*, *crol*, *EcR*, *Cbs* changed only in the *E(z)/w* males. It is also worth noting that in *w/w* males the interventions caused a slight increase in the *Sod2* gene expression, while in *E(z)* mutants no changes were observed.

## Discussion

In the current study, using a combination of several geroprotective interventions, we managed to increase the lifespan of flies by more than 2 times, which is significantly more than using each intervention separately. This result is most likely associated with the synergistic effect of interventions that led to a global metabolic network reorganization and ultimately to beneficially affected lifespan through the modulation of several molecular signaling pathways at once[20].

It is well known that lipid homeostasis is impaired during aging in many eukaryotic organisms from worms to humans. Lipid analysis revealed that the main factors affecting total lipid contents are sex, age, and genotype, but not the combination of

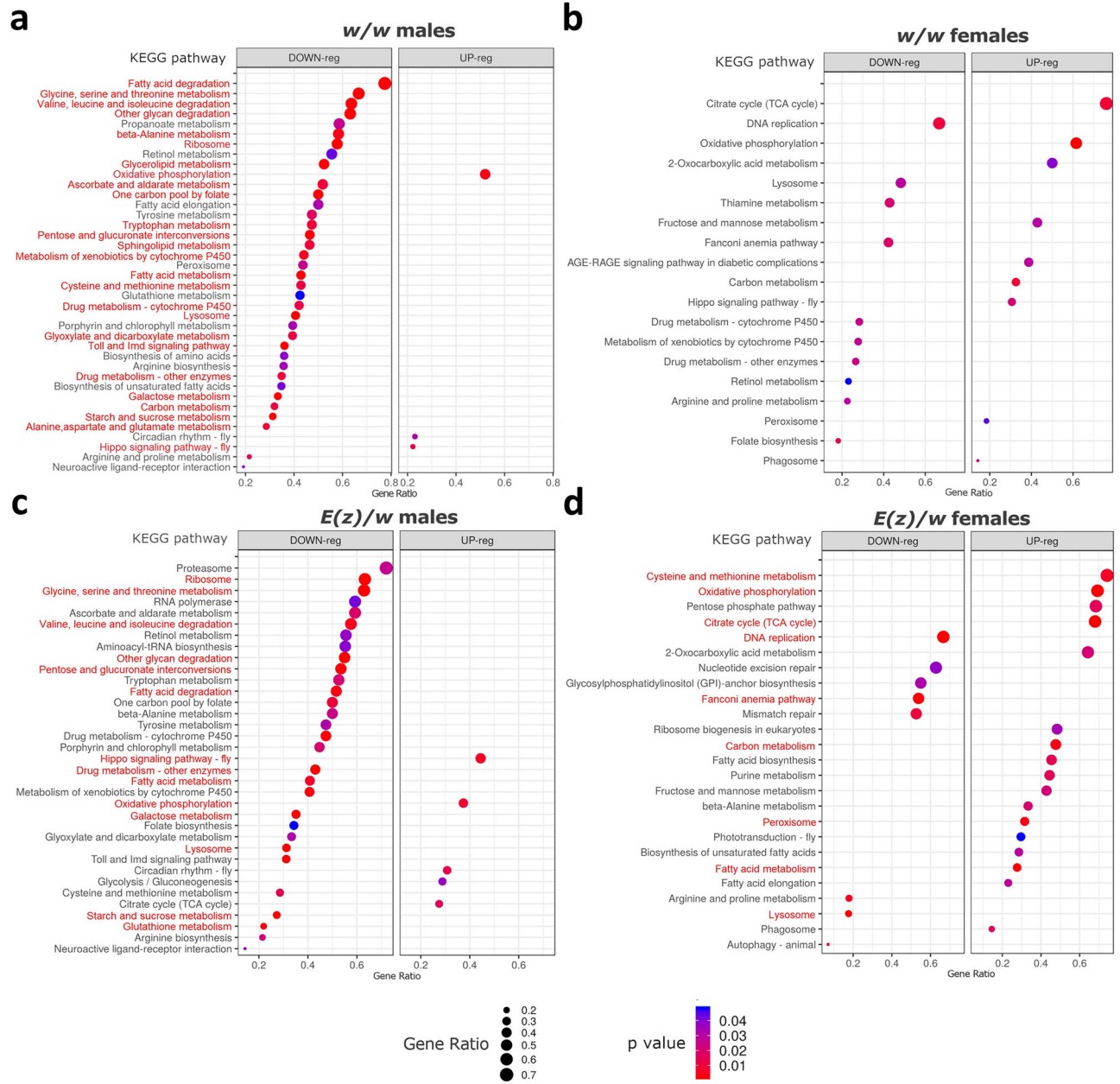

**Fig. 6 Dotplots showing the results of KEGG pathways enrichment analyses performed for DE genes (either up-regulated or down-regulated) associated with exposure to a combination of anti-aging interventions (dietary restriction (DR); co-administration of berberine, fucoxanthin, and rapamycin (3G); living in constant darkness (DD) and low temperature (18 °C) conditions) in 5-day-old *D. melanogaster*. a** *w/w* males, **b** *w/w* females, **c** *E(z)/w* males, **d** *E(z)/w* females. The X-axis and dot size indicate gene ratio (the number of DEGs involved in the KEGG pathway divided by a total number of genes that are annotated as participants of this pathway). Pathways passed the FDR < 0.05 threshold are highlighted in red.

ambient conditions. Thus the lifespan-extending effect of conditions combination seems to be not associated with total lipid contents.

At the same time at the level of gene expression, we found modulation of pathways such as "Fatty acid metabolism", including "Fatty acid elongation", "Fatty acid degradation", "Biosynthesis of unsaturated fatty acids" and "Retinol metabolism", which was reported to be decreased during DR in humans[57]. These pathways may be involved in lipid composition changes, however, we did not study lipid profiles.

The trend of increase in total lipid levels in the experimental *w/ w* and *E(z)/w* flies persisted up to 100 days, then reduced towards the end of life. Interestingly, such a pattern of changes in lipid profiles is consistent with the recently shown age-associated

biphasic alteration in metabolic activity that is apparently associated with mitochondrial function[58]. CR has previously been shown to increase mitochondrial biogenesis[59]. Mitochondria isolated from calorie-restricted yeast showed increased respiration, an enhanced antioxidant defense system, and increased ROS levels, that as suggested by the authors might be associated with the mitohormetic effect[60].

According to our data, a combination of geroprotectors led to an increase in the expression of genes involved in oxidative phosphorylation and the TCA cycle (tricarboxylic acid cycle). Probably due to this, the observed subsequent age-related decrease in the activity of these pathways in the experimental flies is less pronounced compared to the decrease in the control flies.

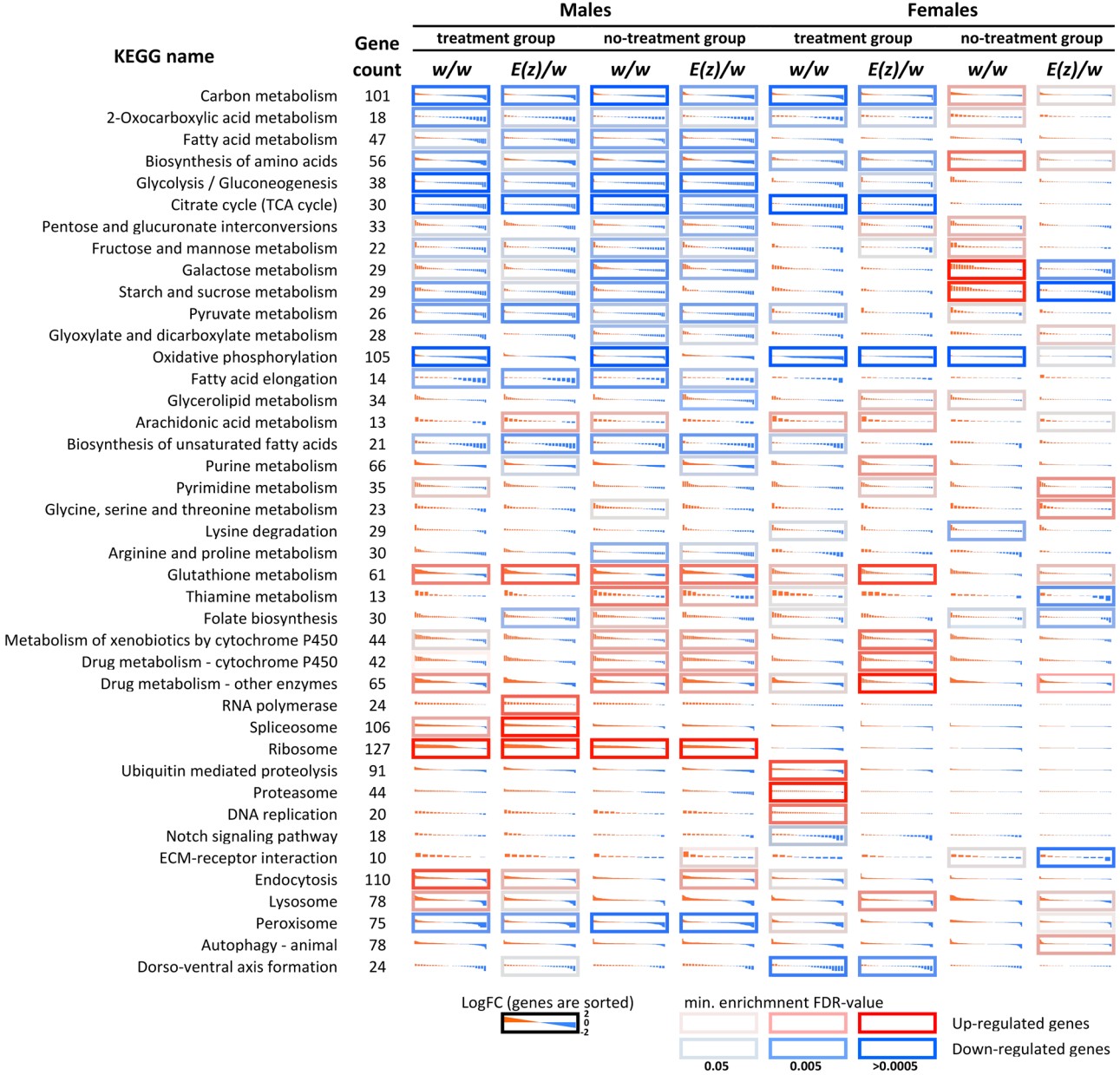

**Fig. 7 Age-dependent changes in the expression profiles measured for genes participating in KEGG pathways in males and females of both genotypes (*w/w* and *E(z)/w*).** Cell borders demonstrate enrichment FDR-value (Fisher's exact test) for a pathway (best value among the enrichments for top-50, 100, etc. DEGs). The red border indicates that a KEGG pathway is predominantly enriched with upregulated genes; the blue border—with downregulated genes. The presented data include genes with average read counts per million (CPM) > 1.

We found that a combination of all studied interventions resulted in a significant decrease in the Glycine N-methyltransferase (*Gnmt*) gene expression, which is somewhat inconsistent with the previously reported data showing that *Gnmt* overexpression extended longevity in *Drosophila*[61]. Gnmt regulates the ratio of S-adenosylmethionine (SAM) to S-adenosylhomocysteine (SAH). SAM is the universal methyl donor required for almost all methyltransferase activity, and, consequently, is important for many physiological processes[62,63]. We also detected a slight decrease in the expression of the *Sam-S* gene encoding the S-adenosyl-methionine synthase enzyme that converts methionine into SAM[64]. More likely, the combination of interventions changed the SAH / SAM ratio, which led to a decrease in *Gnmt* expression[65]. It was previously shown that knockdown of *sams-1* in *Caenorhabditis elegans* resulted in lifespan extension[66],

while the *sams-1* overexpression partially suppressed lifespan extension by DR[67].

We also investigated retrotransposon activity, as it is one of the sources of age-dependent and exogenously induced genomic instability that is involved in the aging process[50,68]. However, in the current study, we failed to reveal a clear relationship between the activity of retrotransposons and lifespan-extending effects of the combination of anti-aging interventions.

Predictably, our results revealed that geroprotective interventions affect the evolutionarily conserved molecular pathways associated with longevity. We found that a combination of interventions modulated the activity of the mTOR signaling pathway. Inhibition of the mTOR signaling pathway leads to improved health and increased lifespan of model organisms ranging from yeast to mammals[69]. Among DEGs, we found an age-dependent decrease in *Tor* gene expression encoding serine/

threonine-protein kinase Tor, which is a key participant of mTOR signaling[70]. The decrease in expression was more pronounced in flies carrying the E(z) mutation. Additionally, only in E(z) mutant flies do we detect a decrease in the Thor gene, eukaryotic translation initiation factor 4E binding protein that is controlled by the product of Tor. In long-lived flies, a slight but statistically significant increase in the expression of the foxo and Akt1 genes was observed, which is consistent with previous studies showing that FoxO is capable to maintain cellular energy homeostasis suppressing the anabolic activity of mTORC1 and simultaneously activating Akt[71].

Autophagy can be induced by a variety of stressors including nutrient and energy stress, oxidative stress, infection, and hypoxia. The maintenance of autophagic activity has been found to contribute to extending longevity[72]. Accumulating evidence indicates that the anti-aging effects of calorie restriction are also due to the induction of autophagy[72]. Surprisingly, treatment by a combination of studied anti-aging interventions resulted in suppression of the autophagy process and lysosomal activity. On the other hand, such differences in the gene expression pattern may be associated with changes in the circadian rhythms of experimental flies which were kept in constant darkness conditions. Recent studies have been indicated that autophagy activity is regulated by both circadian and nutritional signals and varies during the day in several tissues, such as the liver, heart, and muscle[73–75]. There are three general types of autophagy: macroautophagy, chaperone-mediated autophagy (CMA) endosomal microautophagy (eMI)[76]. CMA and eMi are selective forms of autophagy involved internalization of substrates directly into the lysosomal lumen via the lysosomal-associated membrane protein 2 A (LAMP2A) or through invagination of the lysosomal or late endosomal membranes[77]. Both two processes require the presence of KFERQ motif to be recognized by the chaperone HSPA8/Hsp70. However, due to the lack of LAMP2A in Drosophila, which is necessary for CMA, it is assumed that Drosophila uses only eMI as an alternative selective autophagic process[78,79]. It is worth noting that we found a significant decrease in Hsp70 gene expression along with a decrease in the expression of heat shock protein genes.

In conclusion, referring to accumulated data on D. melanogaster longevity this is the first report on the increase of maximum flies' lifespan to more than 200 days (120% increase) under the influence of a combination of geroprotective interventions, in particular, low ambient temperature, dietary restriction, photodeprivation, a combination of rapamycin, berberine, and fucoxanthin. In addition, we observed a decrease in locomotor activity that is apparently a consequence of low temperature and constant darkness conditions. At the same time, we failed to reveal strong associations between lifespan effects and stress resistance, total lipid contents, or retrotransposons activity. This reflects the complex relationship of the studied parameters with genotype, sex, age, and experimental conditions. Probably, synergistic or additive effects of selected interventions triggered a wide number of responses at gene expression levels involved in the processes of autophagy, immune response, epigenetic landscape regulation, energy, and nutrient signaling, which led to a delay in aging and an increase in lifespan. Taking into account evolutionary conservation in a gene regulatory network of the longevity that integrates certain signaling pathways we believe our results can be useful for the development of novel anti-aging approaches with therapeutic potential.

## Methods

**Drosophila melanogaster strains**. The crossing scheme for obtaining w/w and E(z)/w flies was described earlier[28]. Briefly, to align the genetic background of parental lines E(z)[731]/TM6C (#24470, Bloomington, USA) and E(z)[+]/TM6C, a series of 8 backcrosses with line w[1118] (#3605, Bloomington, USA) was performed. All experimental procedures were performed on the F[1] offspring from crosses of w[1118] females with E(z)[+]/TM6C or E(z)[731]/TM6C males. The E(z)[+]/w[1118] (hereinafter referred to as w/w) was used as a control line with a w[1118] genetic background which has a normal lifespan, and E(z)[731]/w[1118] (hereinafter referred to as E(z)/w) – as a long-lived line.

**Maintenance of Drosophila**. The flies were nourished at 25 °C, 60% relative humidity, 12 h day/night cycle. A rearing medium containing water – 1 L, corn flour – 76.6 g, yeast – 32.1 g, agar-agar – 9.3 g, glucose – 63.2 g, sucrose – 31.6 g, and CaCl$_2$ – 0.7 g was used[80]. The rearing medium was supplemented with 5 ml of propionic acid (#P1386, Sigma-Aldrich) and 10 ml of 10% nipagin in 95% ethanol (#H5501, Sigma-Aldrich) to prevent microbial growth. Within 24 h after imago hatching, flies were randomly distributed between the control and experimental groups using carbon dioxide (CO$_2$) anesthesia apparatus "Benchtop Flowbuddy Complete" (#59-122BCU, Genesee Scientific, USA). Adult flies from control and experimental variants were maintained in different conditions throughout life. These differences were related to diet, treatment with geroprotective substances, ambient temperature, and lighting conditions. All variants were kept at a relative humidity of 60%. Constant climate chambers Binder KBF720-ICH (Binder, Germany) were used to maintain stable conditions.

**Dietary restriction**. Control and experimental flies were kept on two different types of food medium, which were isocaloric (~0.77 calories/gm medium) and consisted of the same components, but different in protein-carbohydrate contents. A food medium containing ~6.9% proteins and ~88.8% carbohydrates were used as a control diet (CD). CD contains water – 1 L, corn flour – 92 g, yeast – 32.1 g, agar-agar – 5.2 g, glucose – 136.9 g. A food medium containing fewer proteins (~4.8%) while more carbohydrates (~91.5%) was used as an experimental dietary restricted (DR) medium. DR contains water – 1 L, corn flour – 92 g, yeast – 17.2 g, agar-agar – 5.2 g, glucose – 147.2 g. The propionic acid (5 ml) and nipagin (10 ml of 10% solution in 95% ethanol) were added in both types of food medium for microbial growth inhibition.

The recipes were adapted from Xia and de Belle[80] since they differently affected the Drosophila lifespan. DR was shown to extend Drosophila longevity compared to CD[80]. This diet has been shown to affect Drosophila longevity through E(z)-mediated epigenetic mechanisms[80,81] and was used to study a possible role of E(z) in diet-mediated longevity prolongation.

**Treatment with drugs**. Experimental flies were kept on a nutrient medium supplemented with a mixture of three different compounds (3G) with previously revealed geroprotective activity, including fucoxanthin[27,82] (#F6932, Sigma-Aldrich, USA), rapamycin[83] (#R0395, Sigma-Aldrich, USA) and berberine[84] (#B3251, Sigma-Aldrich, USA). The treatment with substances was carried out according to the previously described method[85], which was adapted from Landis, et al.[86]. Briefly, 30 μL of drug stock solutions were applied to the surface of a cooled and solidified nutrient medium and dried under a fan for 30 min. All stock solutions were prepared with ethanol. In the control variants, 30 μL of ethanol was applied to the surface of the medium. We used the following concentrations of drugs in stock solutions: fucoxanthin – 1 μM, rapamycin – 50 μM, berberine – 1 mM. Blue food dye (FD&C Blue Dye no. 1) which was used as a tracer of drug dilution in food media, revealed that within three days the 30 μL dye solution was absorbed into the top ~0.6 of food media, corresponding to a 20-fold dilution of the stock solutions. Thus, a final concentration of drugs in the medium was: fucoxanthin – 0.05 μM, rapamycin – 2.5 μM, berberine – 50 μM. To avoid the adverse effects, the selected drug concentrations were close to the minimal range of concentrations at which geroprotective effects were manifested in different models for treatment with fucoxanthin (0.15–150 μM)[87], rapamycin (0.005–5 μM)[22], berberine (5–200 μM)[25,88]. Treatment with drugs was started from the first day of imago life and continued through the lifespan.

**Reduced temperature**. Experimental flies were kept at a constant ambient temperature of +18 °C throughout imago life (18 °C). Control individuals were kept at a constant ambient temperature of +25 °C.

**Light regime**. To assess the effect of constant darkness on lifespan, the experimental individuals were maintained under continuous dark (DD), and the controls were kept in the 12 h light / 12 h dark (LD) conditions.

**Lifespan analysis**. The flies were obtained from synchronously layed eggs and collected within 24 h after imago hatching using CO$_2$ anesthesia. Males and non-virgin females were placed separately in "Drosophila Vials, Narrow (PP)" (#32-120, Genesee Scientific, USA) (with 30 individuals in each vial) filled with control or experimental food mediums. Dead flies were counted every 24 h. Flies were transferred to fresh vials twice a week without anesthesia.

**Locomotor activity analysis**. Age-dependent changes in locomotor activity were assessed in flies at 1–10 weeks of age using the LAM25 Locomotor Activity

Monitor (TriKinetics, USA). The analysis was conducted using "*Drosophila* Vials, Narrow (KR)" (#32-118, Genesee Scientific, USA) with improved transparency. The data from the activity monitor (the number of actuation of the infrared motion sensors) were recorded for 24 h and presented as the average daily activity per 1 individual.

**Analysis of stress resistance**. The procedure for analysis of stress resistance was described earlier[28]. Briefly, females and males were selected using $CO_2$ anesthesia and maintained under control or experimental conditions until 10, 30, and 45 days of age. Under the experimental conditions, flies were exposed to a combination of dietary restriction (DR), a mixture of rapamycin, berberine, and fucoxanthin (3G), low temperature (+18 °C), and constant darkness (DD). To assess the resistance to oxidative stress individual flies were placed into transparent glass tubes (5 mm outside diameter × 65 mm length) and analyzed using DAM2 *Drosophila* Activity Monitor (Trikinetics, USA), which constantly monitors locomotor activity via an infrared-light barrier. The food mixture consisted of 2% agar and 5% sucrose with the addition of 20 mM oxidative stress inductor paraquat (#856177, Sigma-Aldrich, USA) was placed into one end of each tube to expose the fly to paraquat constantly over the course of a multi-day experiment. Flies were kept in the 12 h light / 12 h dark conditions, at a temperature of +25 °C, and 60% relative humidity. Locomotor activity data from individual flies were recorded at 60 minute periods and analyzed. The time of death was determined by the complete absence of movement. Based on the obtained data, the median and maximum survival time (50 and 90 percentiles, respectively) were calculated and survival curves were drawn.

**Determination of total lipids**. The colorimetric method of Van Handel[89] was modified for *Drosophila* as described in Eremina and Gruntenko[90] to determine the total lipids in control and experimental variants. Each fly was homogenized in 100 μl of the chloroform-methanol mixture (V/V) and was shaken for 10 min. The supernatant was carefully collected with a pipette and heated at 90 °C in micro-thermostat M-208 (Bis-N, Russia) until the solvent completely evaporated. Next, 10 μl of sulfuric acid (93.5–95.6%) was added to the supernatant and it was heated at the same temperature for 2 more min. Then a sample was cooled on ice and phosphovaniline reagent was added to it up to a total volume of 1 ml. Fresh phosphovanilline reagent had been prepared before each experiment by dissolving 120 mg of vanillin in 20 ml of hot water. Subsequently, 80 ml of phosphoric acid (85%) was added to the mixture to a final concentration of 1.2 mg/ml and the resulting reagent was stored in the dark until use. To obtain the sulfopho-sphovanilline reaction that serves for lipid quantification, the sample was incubated for 15 min at room temperature in the dark until the appearance of a pink col-oration, which was stable for 1 hour. The optical density of the obtained reaction product was measured with a SmartSpec™ Plus spectrophotometer (Bio-Rad, USA) at 525 nm against a "blank" sample containing only phosphovaniline reagent. Calibration curves were done with the use of refined sunflower oil dissolved in chloroform.

**Analysis of retrotransposon expression level**. The analysis of retrotransposon activity was performed by reverse-transcription quantitative PCR (qRT-PCR). Flies were maintained under control or experimental conditions until 10 and 30 days of age. Under the experimental conditions, flies were exposed to a combination of dietary restriction (DR), a mixture of rapamycin, berberine, and fucoxanthin (3G), low temperature (18 °C), and constant darkness (DD). The whole body flies (20 males and 10 females per each variant) were placed in Aurum Total RNA Lysis Solution (Bio-Rad, USA). RNA was isolated using the Aurum Total RNA Mini Kit (Bio-Rad, USA) according to the manufacturer's instructions. cDNA synthesis was performed using the iScript cDNA Synthesis Kit (Bio-Rad, USA) according to the manufacturer's instructions. For PCR, reaction mixtures were prepared based on qPCRmix-HS SYBR (Evrogen, Russia) with the addition of primers (Syntol, Russia) for the housekeeping genes or retrotransposons (Supplementary Table 17) and cDNA samples. Amplification for each variant of the experiment was carried out in separate tubes in a CFX96 real-time PCR detection system (Bio-Rad, USA) according to the following program: (1) 95 °C for 30 s, (2) 95 °C for 10 s, (3) 60 °C for 30 s, (4) steps 2–3 were repeated 49 times.

**Statistics and reproducibility**. Differences between survival curves were analyzed by the log-rank test[91]. The 50th (median lifespan), 90th, and 100th (maximum lifespan) percentiles of lifespan were estimated. Fisher's exact test[92] was applied for testing the differences in median lifespan and in the 90th percentile of lifespan according to Wang et al.[93] recommendations. Bonferroni correction was used in all multiple comparisons. Cox proportional-hazard regression model was utilized to evaluate the effect of diet, geroprotectors, temperature, lighting, sex, and genotype on survival[94]. Lifespan experiments were performed at least twice, three to five vials (30 flies per vial) were used in each replication of the experiment.

Two-way ANOVA was used to analyze differences in locomotor activities, depending on age and treatment. If a significant difference ($p < 0.05$) was found, the post-hoc pairwise comparisons were performed using Tukey's Honestly Significant Difference (HSD) tests. Spearman's correlation coefficients (r)[95] were used to measure the degree of association between the level of locomotor activity and experimental factors. Bayesian ANOVA[96] was used to reveal factors that make the

greatest contribution to locomotor activity. For the locomotor activity analysis, 5 vials were used, each of which was considered as a replication of the experiment. Each vial contained 10 flies.

Four-way ANOVA was used to analyze differences in resistance to oxidative stress depending on sex, age, a combination of experimental conditions, and *E(z)/w* genotype. The post hoc analysis was performed using Bonferroni correction. Bayesian analysis was used for ANOVA models comparison[96]. For each experimental variant, 32 males and females were analyzed. Each experiment was repeated 3–5 times. A total of 96–160 males and females were analyzed for each variant.

The data of total lipid content were analyzed using a multi-factor ANOVA. Bayesian factor $BF_{10}$ was used to quantify evidence for the best alternative hypothesis relative to the null model, when needed[97].

Expression levels of retrotransposons (target genes) were calculated relative to the expression of *β-Tubulin at 56D*, *RpL32*, and *EF1α* (reference genes). The ΔCq = Cq (Target gene)—Cq (Reference genes), where Cq—quantification cycle[98]. Bayesian ANOVA was used to identify common factors that affect the expression level of different retrotransposons[96]. Four-way (factored by sex, age, genotype, and experimental conditions) or three-way (factored by age, genotype, and experimental conditions) analysis of variance with covariate (ANCOVA) was applied to the qRT-PCR data. The data were normalized by using the covariates which included means of ΔCq values in control males (four- and three-way ANOVA) or females (three-way ANOVA) at the age of 10-days. At least 15% differences between ΔCq values were considered statistically significant. The multiple comparisons (post-hoc testing) were taken into account using Bonferroni correction. The value of Cq was taken from the CFX Manager software (Bio-Rad, USA).

Statistical data processing was performed using the CFX96 Software (BioRad, USA), Statistica Ultimate Academic (version 13.3, TIBCO Software, USA), OASIS 2 (Online application for survival analysis)[99], JASP (version 0.16, JASP team, Netherlands)[100], and Microsoft Excel 2019 (Microsoft, USA).

**Preparation of cDNA libraries and sequencing**. Male and female flies from control and experimental groups (50 whole fly bodies of each sex for each variant) were separately collected under light $CO_2$ anesthesia, snap-frozen in liquid nitro-gen, and stored at −80 °C. Each experiment was replicated three times. We have abbreviated the title of the samples for easy reference (for example, M1Ez5E means Male, 1st replicate, E(z) mutant, 5 days, Experimental (treated)). All analyzed variants are presented in Supplementary Table 16. For transcriptomic analysis, total RNA from 30 flies (10 flies per replicate) was isolated using QIAzol Lysis Reagent (Qiagen, Netherlands) with the isopropanol precipitation. Then the samples were treated with DNase I (Promega, USA). The RNA concentration was determined using a Qubit 2.0 Fluorometer (Thermo Fisher Scientific, USA). The integrity of isolated RNA was evaluated using an Agilent 2100 Bioanalyzer (Agilent Technologies, USA). For subsequent library preparation samples with RIN (RNA Integrity Number) >8 were used. Libraries were prepared from 1 mg of total RNA using a TruSeq RNA Library Prep Kit v2 (Illumina, USA) according to the man-ufacturer's protocol. cDNA libraries were normalized to 4 nM, pooled together, and sequenced with 75 bp single-end reads on the NextSeq500 System (Illumina, USA). The sequencing data are available at the NCBI Sequence Read Archive (PRJNA757594).

**The analysis of transcriptome sequencing data**. The analysis of RNA-Seq data was performed as described earlier[28]. First, sequencing quality was evaluated using FastQC 0.11.9 and then the reads were trimmed with Trimmomatic 0.39. The contamination with bacterial RNAs and the efficacy of polyA selection were evaluated with mapping reads to bacterial genomes (all strains that had been submitted to NCBI Genome database up to 2015) and *D. melanogaster* rRNA genes, correspondingly, with BWA 0.7.17[101]. Typically, rRNA ratio was 0.7–2.5%. Bacterial contamination did not exceed 1.4%.

Next, reads were mapped to the *D. melanogaster* genome (BDGP6, Ensembl release 90) with STAR 2.7[102]. About 90–95% of reads were uniquely mapped. In order to ensure the absence of the impact of possible RNA integrity differences, we evaluated 5'−3' transcript read coverage profiles using modified geneBody_coverage.py script from the RSeQC 3.0.1 toolkit[103]. Read counts per gene were evaluated with featureCounts tool from the Subread 1.6.0 package[104].

Finally, the differential expression analysis using the derived read counts was done with edgeR 3.28.1[105]. The genes were pre-filtered by the expression level: read counts per million, CPM > 1.0 for at least 50% samples of the smallest group under comparison. Then, after TMM normalization, we performed inter-group comparisons and regression analysis with generalized linear models using both the likelihood ratio (LR) test and quasi-likelihood F-test (QLF test). The analyses were made separately for male and female organisms, *w/w* and *E(z)/w* genotypes, treated and non-treated conditions. Next, we performed Gene Ontology, KEGG, Reactome enrichment tests for the lists of top-50, 100, 200, 500, and 1000 differentially expressed genes (DEGs), which were ranged by increasing *p*-value (QLF test; only genes with $p < 0.01$ and |LogFC| > 0.3 were included in the analysis). The results of the enrichment analyses significantly differ for various top DEGs lists and complement each other. For the enrichment tests, topGO 2.38 and clusterProfiler 3.14.3 Bioconductor packages were used[106]. KEGG pathways visualization was

performed using the modification of pathview 1.12 package[107] as described earlier[108].

**Reporting Summary**. Further information on research design is available in the Nature Research Reporting Summary linked to this article.

## Data availability

The derived RNA-Seq data are available in the NCBI Sequence Read Archive (BioProject ID PRJNA757594). All source data underlying the graphs and charts presented in the main figures available in Supplementary Data 2.

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

## Acknowledgements

We thank the Center for Precision Genome Editing and Genetic Technologies for Biomedicine, EIMB RAS for providing the computing power and sequencing data analysis. The sequencing was performed using the equipment of EIMB RAS "Genome" center (http://www.eimb.ru/ru1/ckp/ccu_genome_c.php). The authors would like to thank Nicolay Sidorov for his assistance in purchasing of research reagents at the initial stage of the project. The study was carried out within the framework of the state task on themes "Genetic and functional studies of the effects of geroprotective interventions on the *Drosophila melanogaster* model", N 122040600022-1. The analyses of total lipids content were performed by N.E.G., M.A.B. & P.N.M. with support from #FWNR-2022-0019 MSHE project.

## Author contributions

Conceptualization, A.A.M. and M.V.S.; Methodology, M.V.S., N.V.Z., Z.G.G., G.S.K., N.E.G. and A.V.K.; Software, M.V.S., N.V.Z., Z.G.G., G.S.K. and P.N.M.; Investigation, M.V.S., N.V.Z., Z.G.G., L.A.K., E.V.S., A.A.G., D.A.G., N.R.P., N.S.U., I.A.S. and M.A.B.; Data Curation, M.V.S., N.V.Z., Z.G.G., G.S.K., P.N.M. and A.V.K.; Writing – Original Draft Preparation, Z.G.G., M.V.S., N.V.Z., G.S.K. and M.A.B.; Writing – Review & Editing, G.S.K. and A.A.M.; Visualization, M.V.S., N.V.Z., Z.G.G., G.S.K. and P.N.M.; Supervision, A.A.M.; Project Administration, M.V.S., Z.G.G., A.V. K. and A.A.M.;

Funding Acquisition, A.A.M. All authors have read and agreed to the published version of the manuscript.

## Competing interests

The authors declare no competing interests.
