## [Peer Review File · Communications Biology]

Reviewers' comments:

Reviewer #1 (Remarks to the Author):

As the world population becomes increasingly older, increasing healthy lifespan becomes important for delaying age-related diseases. Various factors have been associated with increased lifespan in a wide variety of species, including dietary restriction (DR) and reduced temperature. This study uses *Drosophila* to assess the effects of four factors previously associated with increased lifespan, jointly and in combination: DR, 18°, constant dark (DD), and joint treatment with three compounds: berberine, fucoxanthin, and rapamycin (3G). The authors find that treatment with all four factors considerably extends lifespan in both sexes and two genetic backgrounds (one long-lived). They then go on to examine effects of this combined treatment on locomotor activity, oxidative stress resistance, total lipids, retrotransposon activity, and the transcriptome, separately for males and females and the two genotypes. Locomotor activity was negatively correlated with increased lifespan, and the other traits had effects that were largely dependent on sex, age and genotype.

Major comments

1. The manuscript is difficult to understand. This is primarily because the Results sections describe what was found, without explaining the experimental set-up first, so the reader has to constantly flip between the description of the results and the Methods describing the experiments.
2. The statistical analyses are not clear and not given. The analysis of the four-factor lifespan experiment should be by four way factorial ANOVAs with cross-classified main effects of DR, 18°, DD, and 3G, separately for males, females and the two genotypes. The figures indicate that this was the experimental design used. It appears that each of the treatment combinations is compared to the control. The more sophisticated design treating lifespan as a quantitative trait will enable us to see which of the main effects are significant and whether there are enhancing or suppressing interactions between the treatment. Just looking at the figure, I wonder whether the 18° treatment accounted for all of the increased lifespan. If so, a three-way factorial analysis fitting main effects of sex, genotype and treatment will highlight context-dependent effects on lifespan.
3. Similarly, for the locomotor, stress resistance and lipid content data, four-way ANOVAs with effects of sex, age, genotype and treatment is appropriate (only for the young and 50 day old flies for lipid content). The retrotransposon data is hard to interpret as presented. Again, separate four way ANOVAs with main effects of sex, age, genotype and treatment for each retrotransposon would help. It is not clear why whole bodies were profiled for retrotransposon activity, as it is expected that different tissues (e.g., brains vs. reproductive tissues) will have different transposition rates with age.
4. I do not see any statistical analyses of the transcriptome data. How was differential expression determined? Separate analyses of males, females for each of the two genotypes and by age, comparing young flies with older flies? Was a false discovery threshold of 0.05 used to declare differential expression? KEGG and Reactome analyses should only use statistically significantly differentially expressed genes, not arbitrarily chosen ranks (top 50, 100, 200 etc.)
5. Raw data are not given.

Reviewer #2 (Remarks to the Author):

In this manuscript, Shaposhnikov et al., examine the cumulative effects on fruit flies of several interventions previously shown individually to extend lifespan. The authors use dietary, husbandry, genetic, and chemical interventions to produce a combined fly lifespan exceeding 200 days. They also perform some tests of hypotheses that have been proposed as drivers of aging, including lipid content, transposons, and stress resistance via oxidative stress. The lifespan extension is

noteworthy, and the variety of conditions tested would seem to be of value to the aging community in particular. In general, the manuscript needs some language editing for clarity in several spots, as it is sometimes difficult to determine what the authors are referring to. Several major issues need to be addressed, and specific concerns/comments about these are below.

1. The DR regimen described in the manuscript does not seem to be canonical DR – there is a shift in terms of the macronutrient content but no caloric restriction. Based on the supplied reference (Xie and de Belle) the DR diet seems to be essentially a mild protein restriction, which nevertheless produced a robust lifespan extension in the original reference (which the authors of this manuscript note in the methods). Two questions about this -

a. What was the rationale for this particular DR regimen? Nutrient dilution, to achieve something closer to a true caloric restriction, can produce significant lifespan extension, but that was not used here.

b. In this manuscript, the DR regimen does not seem to produce any lifespan extension, and even produces a slightly higher hazard ratio in the wild type flies at least, contradicting the provided reference (Xie and de Belle). Do the authors have an explanation for this?

2. “Despite the molecular mechanisms underlying this effect remain largely unknown, experimental data suggest that the main role play changes in the expression level of genes affecting the rate of aging.” It is not clear what the authors are saying with this sentence (one of the examples of needed editing for clarity). Are they suggesting that expression level changes to the example genes they provide (TRP7, etc) are responsible for the lifespan effects of lowered temperature? Or just that lowered temperature leads to gene expression changes in these example genes, some of which have been linked to lifespan?

3. “Obviously, the decrease in locomotor activity was mainly related to low-temperature conditions and constant darkness.” This statement is not supported by any evidence or literature citations. I can appreciate that, intuitively, it makes sense; however, the authors presumably have the data from all of the individual conditions that they tested, and could describe if any individual one, including darkness and temperature, produce any locomotor defects. They should do so, or change the statement.

4. “...suggesting reduced or slowed senescence...” This is also not supported by any evidence, as the authors did not measure senescence.

5. “Probably the influence of this combination of interventions on the transcriptome of *D. melanogaster* with *E(z)* mutation becomes stronger with age compared to wild-type flies.” This statement doesn’t seem supported by the experiments performed...counting the number of genes that pass certain arbitrary thresholds does not necessarily indicate a stronger level of influence over the transcriptome, and there are also no intermediate time point analyses with which to compare to determine how this effect is changing with age. It’s possible that the pattern of transcriptome changes is similar in wild-type and *E(z)* flies of the same biological age but not chronological age, hence the observed discrepancy.

6. “Tor...controls the aging process” This is also not supported by the literature or the data in this manuscript. Tor is certainly important for aging, but marking it down as the thing that controls aging is not correct, nor supported by the evidence.

7. “several key mechanisms controlling the aging processes...” (Line 261). Again, I don’t think ‘control’ or ‘controlling’ when describing factors that influence aging is appropriate. The degree to which any of those pathways truly dictates the aging process is not well established.

a. Another example – “Tor regulates energy homeostasis and controls the aging process” (Line 287)

8. I don’t see any statement regarding data availability, particularly for the transcriptome dataset. It is not part of the supplemental data, so I cannot evaluate it. This should be deposited in a publicly accessible repository.

Dear Editor and Reviewers:

Thank you for giving us the opportunity to submit a revised draft of the manuscript "Molecular mechanisms of exceptional lifespan increase of *Drosophila melanogaster* with different genotypes after combinations of pro-longevity interventions". We appreciate the time and effort that you and the reviewers have dedicated to providing your valuable feedback on our manuscript. We have studied comments carefully and have made corrections as marked in the revised manuscript, which we sincerely hope will meet with your approval.

Below is our point-by-point detailed response to the reviewers.

Response to the reviewers' comments:

Reviewer #1 (Remarks to the Author):

As the world population becomes increasingly older, increasing healthy lifespan becomes important for delaying age-related diseases. Various factors have been associated with increased lifespan in a wide variety of species, including dietary restriction (DR) and reduced temperature. This study uses *Drosophila* to assess the effects of four factors previously associated with increased lifespan, jointly and in combination: DR, 18°, constant dark (DD), and joint treatment with three compounds: berberine, fucoxanthin, and rapamycin (3G). The authors find that treatment with all four factors considerably extends lifespan in both sexes and two genetic backgrounds (one long-lived). They then go on to examine effects of this combined treatment on locomotor activity, oxidative stress resistance, total lipids, retrotransposon activity, and the transcriptome, separately for males and females and the two genotypes. Locomotor activity was negatively correlated with increased lifespan, and the other traits had effects that were largely dependent on sex, age and genotype.

Major comments

1. The manuscript is difficult to understand. This is primarily because the Results sections describe what was found, without explaining the experimental set-up first, so the reader has to constantly flip between the description of the results and the Methods describing the experiments.

[Response]

This comment has been taken into account. We have included short descriptions that explain the experimental setup. The following changes were made in the Results sections:

Lifespan. The effects of different conditions (maintaining in the dark (DD), low temperature (18°C), exposure to geroprotectors (3G), dietary restriction (DR)) with potential geroprotective activities and its possible combinations on the lifespan of the *w/w* line and long-lived mutants *E(z)/w* were analyzed. ...

Locomotor activity. The effects of condition combinations (DR, 3G, 18°C, DD) on age-dependent dynamics of locomotor activity of both *w/w* and *E(z)/w* flies were analyzed as the indicators of healthspan (Avanesian et al., 2010). ...

Transcriptome analysis. Next, we performed RNA-Seq to analyze the effects of DR, 3G, 18°C, DD combination on gene expression profiles of *w/w* and *E(z)/w* flies. To assess the transcriptomic dynamics during the aging of flies we used four age groups: females and males of both genotypes at the age of 5, 50, 100, and 150 days old. All analyzed variants are presented in Supplementary Table S10. According to differential expression analysis the combination of interventions caused changes in expression of 31 genes in 50-day-old *w/w* females (Fold change(FC) > 2, LogCPM > 1, p<0.05, no genes passed FDR< 0.05) and 772 genes in 50-day-old *w/w* males (among them 745 genes with FDR < 0.05). At the same time, the changes in long-lived *E(z)/w* flies of the same age were more dramatic: 556 (523 with FDR < 0.05) differentially expressed genes (DEGs) in females and 1396 DEGs (1372 with FDR < 0.05) in males were detected. In 5-day-old flies, the number of DEGs was approximately the same in both genotypes: 388 (377 with FDR < 0.05) DEGs in *w/w* females, 579 (574 with FDR < 0.05) DEGs in *w/w* males, 502 (487 with FDR < 0.05) DEGs in *E(z)/w* females, 485 (479 with FDR < 0.05) DEGs in *E(z)/w* males.

2. The statistical analyses are not clear and not given. The analysis of the four-factor lifespan experiment should be by four-way factorial ANOVAs with cross-classified main effects of DR, 18°, DD, and 3G, separately for males, females, and the two genotypes. The figures indicate that this was the experimental design used. It appears that each of the treatment combinations is compared to the control. The more sophisticated design treating lifespan as a quantitative trait will enable us to see which of the main effects are significant and whether there are enhancing or suppressing interactions between the treatment. Just looking at the figure, I wonder whether the 18° treatment accounted for all of the increased lifespans. If so, a three-way factorial analysis fitting the main effects of sex, genotype, and treatment will highlight context-dependent effects on lifespan.

[Response]

Thanks for your suggestions and sorry for these flaws in the statistical analyses.

It should be noted that there are certain aspects of survival analysis data, such as censoring and non-normality, that generate great difficulty when trying to analyze the

data using traditional statistical models including traditional ANOVA (Lee, E.T. & Wang, J.W. Statistical methods for survival data analysis, 513 p. (J. Wiley, New York, 2003) http://www.ru.ac.bd/wp-content/uploads/sites/25/2019/03/403_05_Lee_Statistical-Methods-for-Survival-Data-Analysis-Third-Edition-Wiley-Series-in-Probab.pdf).

Considering the difficulties of ANOVA using, Cox regression analysis was performed to investigate the relationship of factors (DD, 18°C, 3G, and DR) and the risk of death through the hazard function separately for males, females, and the two genotypes. To address whether the 18° treatment accounted for all positive effects on lifespan, Cox regression analysis was performed separately for 25°C and 18°C temperature variants. This analysis revealed the condition-dependent lifespan effects of DD, DR, 3G, sex, and genotype.

All these changes are presented in the Results and Methods sections.

3. Similarly, for the locomotor [see Response 1], stress resistance [see Response 2], and lipid content data [see Response 3], four-way ANOVAs with effects of sex, age, genotype, and treatment is appropriate (only for the young and 50-day old flies for lipid content). The retrotransposon data is hard to interpret as presented [see Response 4]. Again, separating four way ANOVAs with main effects of sex, age, genotype and treatment for each retrotransposon would help. It is not clear why whole bodies were profiled for retrotransposon activity, as it is expected that different tissues (e.g., brains vs. reproductive tissues) will have different transposition rates with age [see Response 5].

[Response]

Thank you for these comments. We have divided the response into several subparagraphs in accordance with the given numbers. All relevant revisions have been included in the manuscript (Results and Methods sections).

[Response 1]

- locomotor activity

Two-way ANOVA was used to analyze differences in locomotor activities, depending on age and treatment. If a significant difference ($p < 0.05$) was found, the post-hoc pairwise comparisons were performed using Tukey's Honestly Significant Difference (HSD) tests. Spearman's correlation coefficients (r) were used to measure the degree of association between the level of locomotor activity and experimental factors. Bayesian ANOVA was used to reveal factors that make the greatest contribution to locomotor activity.

[Response 2]

- stress resistance

Four-way ANOVA was used to analyze differences in resistance to oxidative stress depending on sex, age, a combination of experimental conditions, and $E(z)/w$ genotype. The post hoc analysis was performed using Bonferroni correction. Bayesian analysis was used for ANOVA models comparison.

[Response 3],

- lipid content data

The data of total lipid content were analyzed using a multi-factor ANOVA. Bayesian factor BF_{10} was used to quantify evidence for the best alternative hypothesis relative to the null model, when needed

[Response 4]

- retrotransposon data

Bayesian ANOVA was used to identify common factors that affect the expression level of different retrotransposons {Rouder, 2016 #137}. Four-way (factored by sex, age, genotype, and experimental conditions) or three-way (factored by age, genotype, and experimental conditions) analysis of variance with covariate (ANCOVA) was applied to the qRT-PCR data. The data were normalized by using the covariates which included means of ΔCq values in control males (four- and three-way ANOVA) or females (three-way ANOVA) at the age of 10-days. At least 15% differences between ΔCq values were considered statistically significant. The multiple comparisons (post-hoc testing) were taken into account using Bonferroni correction.

[Response 5]

-It is not clear why whole bodies were profiled for retrotransposon activity

Thank you for this comment. We included the explanation in the Results section:

“Taking into account that the combinations of used factors and $E(z)$ mutation, may target different tissues {Piper, 2018 #148}{Deleris, 2021 #145}{Chaouch, 2021 #146} including the nervous system, intestine, and fat body for which age-dependent activation of retrotransposons was shown {Li, 2013 #92}{Treiber, 2020 #150}{Chen, 2016 #90}{Wood, 2016 #151}{Sousa-Victor, 2017 #152} the level of transpositions was evaluated in fly whole bodies.”

4. I do not see any statistical analyses of the transcriptome data. How was differential expression determined? Separate analyses of males, females for each of the two genotypes and by age, comparing young flies with older flies? Was a false discovery threshold of 0.05 used to declare differential expression? KEGG and Reactome analyses should only use statistically significantly differentially expressed genes, not

arbitrarily chosen ranks (top 50, 100, 200 etc.)

[Response]

Differential expression analysis was performed using the edgeR package. Likelihood ratio test (LR) and quasi-likelihood F-test (QLF) were applied to assess differential expression. The analyses were made separately for male and female organisms, w/w and E(z)/w genotypes, treated and non-treated conditions.

When the FDR < 0.05 threshold was applied, too few genes had passed this threshold. We decided to use less stringent thresholds, $p < 0.05$ in order to retrieve more information.

We performed Gene Ontology, KEGG, Reactome enrichment tests for the lists of top-50, 100, 200, 500, and 1000 differentially expressed genes (DEGs), which were ranged by increasing p-value (QLF test). Only genes with $p < 0.01$ and $|\text{LogFC}| > 0.3$ were included in the analysis. The results of the enrichment analyses significantly differ for various top DEGs lists and complement each other.

The changes have been made in the Methods.

5. Raw data are not given.

The RNA-Seq data was uploaded to NCBI SRA (BioProject ID PRJNA757594). The information has been added to the manuscript. You also can review the data by following the link

<https://dataview.ncbi.nlm.nih.gov/object/PRJNA757594?reviewer=oatocq9hcavcoctcnkhei3s7b5>

Reviewer #2 (Remarks to the Author):

In this manuscript, Shaposhnikov et al., examine the cumulative effects on fruit flies of several interventions previously shown individually to extend lifespan. The authors use dietary, husbandry, genetic, and chemical interventions to produce a combined fly lifespan exceeding 200 days. They also perform some tests of hypotheses that have been proposed as drivers of aging, including lipid content, transposons, and stress resistance via oxidative stress. The lifespan extension is noteworthy, and the variety of conditions tested would seem to be of value to the aging community in particular. In general, the manuscript needs some language editing for clarity in several spots, as it is sometimes difficult to determine what the authors are referring to. Several major issues

need to be addressed, and specific concerns/comments about these are below.

1. The DR regimen described in the manuscript does not seem to be canonical DR – there is a shift in terms of the macronutrient content but no caloric restriction. Based on the supplied reference (Xie and de Belle) the DR diet seems to be essentially a mild protein restriction, which nevertheless produced a robust lifespan extension in the original reference (which the authors of this manuscript note in the methods). Two questions about this -

a. What was the rationale for this particular DR regimen? Nutrient dilution, to achieve something closer to a true caloric restriction, can produce significant lifespan extension, but that was not used here.

[Response]

We included the explanation in the Methods section:

“This diet has been shown to affect *Drosophila* longevity through *E(z)*-mediated epigenetic mechanisms {Xia, 2016 #34}{Xia, 2016 #33} and was used to study a possible role of *E(z)* in diet-mediated longevity prolongation.”

b. In this manuscript, the DR regimen does not seem to produce any lifespan extension, and even produces a slightly higher hazard ratio in the wild type flies at least, contradicting the provided reference (Xie and de Belle). Do the authors have an explanation for this?

[Response]

We included the explanation in the results section:

“The positive, negative, or neutral effect of dietary and pharmacological interventions may be associated with the general nature of hormetic action of dietary proteins {McCracken, 2020 #125} or anti-aging drug {Shaposhnikov, 2018 #139} concentrations in the nutrition medium, and maybe substantially modified by sex, genotype, and other environmental factors {Jin, 2020 #135}{Lucanic, 2017 #126}.”

2. “Despite the molecular mechanisms underlying this effect remain largely unknown, experimental data suggest that the main role play changes in the expression level of genes affecting the rate of aging.” It is not clear what the authors are saying with this sentence (one of the examples of needed editing for clarity). Are they suggesting that expression level changes to the example genes they provide (TRP7, etc) are

responsible for the lifespan effects of lowered temperature? Or just that lowered temperature leads to gene expression changes in these example genes, some of which have been linked to lifespan?

[Response]

The sentence has been rephrased for clarity.

3. “Obviously, the decrease in locomotor activity was mainly related to low-temperature conditions and constant darkness.” This statement is not supported by any evidence or literature citations. I can appreciate that, intuitively, it makes sense; however, the authors presumably have the data from all of the individual conditions that they tested, and could describe if any individual one, including darkness and temperature, produce any locomotor defects. They should do so, or change the statement.

[Response]

This comment has been taken into account. All relevant revisions have been included in the manuscript (Results and Methods sections).

To address which of the used interventions make the greatest contribution to the level of locomotor activity, we analyzed the motility of flies under various combinations of experimental factors (Supplementary Fig. S2). Spearman's correlation analysis (Supplementary Table S5) revealed moderate (-0.476 and -0.610) but very highly significant ($p < 1 \times 10^{-17}$) associations between the level of locomotor activity and two conditions (18°C and DD, respectively). Evaluation of the contribution of low ambient temperature (18°C) and absence of light (DD) to the locomotor activity was performed using Bayesian ANOVA, taking into account differences in sex, age, and genotype. The model comparison revealed the most probable contribution of factor combination: GT + DD + 18°C (Supplementary Table S6).

The analysis of covariance (ANCOVA) {Maher, 2013 #133} was used to estimate the relative strength of the contribution of DD and 18°C versus sex, age, and genotype (Supplementary Table S7). Partial eta-squared values suggest that the main effect on locomotor activity is associated with maintaining in the dark (DD: $\eta^2 = 0.188$, $p < 0.001$) and, to a lesser extent, genotype (GT: $\eta^2 = 0.007$, $p < 0.001$) and low ambient temperature (18°C: $\eta^2 = 0.006$, $p < 0.001$). At the same time, effect size of sex (SEX: $\eta^2 = 0.003$, $p < 0.05$) and age (AGE: $\eta^2 = 0.002$, $p < 0.05$) were minimal.

4. “...suggesting reduced or slowed senescence...” This is also not supported by any evidence, as the authors did not measure senescence.

[Response]

This statement was deleted.

5. “Probably the influence of this combination of interventions on the transcriptome of *D. melanogaster* with E(z) mutation becomes stronger with age compared to wild-type flies.” This statement doesn’t seem supported by the experiments performed...counting the number of genes that pass certain arbitrary thresholds does not necessarily indicate a stronger level of influence over the transcriptome, and there are also no intermediate time point analyses with which to compare to determine how this effect is changing with age. It’s possible that the pattern of transcriptome changes is similar in wild-type and E(z) flies of the same biological age but not chronological age, hence the observed discrepancy

[Response]

Perhaps you are right, and these conclusions are hasty. We have removed these sentences from the text.

6. “Tor...controls the aging process” This is also not supported by the literature or the data in this manuscript. Tor is certainly important for aging, but marking it down as the thing that controls aging is not correct, nor supported by the evidence.

[Response]

This statement have been corrected:

“As a participant of the mTOR pathway, Tor plays one of the key roles in maintaining energy homeostasis and its modulation influences longevity and aging (Chang, Neufeld, 2009).”

7. “several key mechanisms controlling the aging processes...” (Line 261). Again, I don’t think ‘control’ or ‘controlling’ when describing factors that influence aging is appropriate. The degree to which any of those pathways truly dictates the aging process is not well established.

a. Another example – “Tor regulates energy homeostasis and controls the aging process” (Line 287)

Such statements/sentences have been rephrased or removed from the manuscript.

8. I don’t see any statement regarding data availability, particularly for the transcriptome dataset. It is not part of the supplemental data, so I cannot evaluate it. This should be deposited in a publicly accessible repository.

[Response]

The data was uploaded to NCBI SRA(BioProject ID PRJNA757594). You can review the data by following the link

<https://dataview.ncbi.nlm.nih.gov/object/PRJNA757594?reviewer=oatocq9hcavcoctcnkhei3s7b5>

The information has been added to the “Data availability” section.

Reviewers' comments:

Reviewer #2 (Remarks to the Author):

For the most part the authors have addressed my previous concerns. However, their edits have raised new ones, including spots that were changed that were not marked.

In the paragraph describing the pathway analysis ("To determine what cellular processes are affected by selected...), the initial submission described the threshold for statistically significant enrichment as pathways passing an FDR of less than 0.05. However, in this resubmitted manuscript, that has been changed to p less than 0.05, suggesting that they used an uncorrected p -value threshold; the figure legend matches this. Using a nominal p -value cutoff for pathway enrichment would not be appropriate. See Wijesooriya et al, PLoS Comp. Biol (2022).

Furthermore, their expanded explanation of the statistical filtering performed for the transcriptome analysis also brings up more questions;

- It is quite surprising that there were no transcripts passing an FDR of 0.05 in the 50-day old females but 745 passing that threshold in the 50-day old males. There seems to be a similar effect on survival from the combination treatment, and based on the survival graphs in figure 1, by 50 days of age the survival of treated vs wt flies seems to be similar in males and females, but the transcript expression patterns are so different that there are ~700 genes passing an FDR of 0.05 in the males but none in the females? This seems like it should be investigated further, especially given that this treatment yields 388 transcripts passing an FDR of 0.05 at 5 days of age.
- The shifting comparisons sometimes get a little confusing – the authors describe the DEG landscape between treated and untreated flies of both genotypes for 50 day old flies and 5 day old flies. But there is no mention of the comparable comparisons for 100 and 150 day old flies. Instead, the authors have shifted to comparing the genotypes. It would be helpful to perhaps have a table with all of the other comparisons, the threshold used for determining differential expression, and the corresponding number of DE genes.

In supplement table S16, the listed criteria are cutoff by the page, but it seems $CPM > 0.1$ and $p < 0.05$; however, in the results text, they list $LogCPM > 1$, $p < 0.05$ for the female 50 day old flies; the methods section suggests $p < 0.01$ and $abs(LogFC) > 0.3$ – which one is the correct threshold? Also, the text states "All analyzed variants are presented in Supplementary Table S16." However, Table S16 only has E(z)/w vs w/w for 100- and 150-day old males and females. It doesn't seem to include the w/w treated vs. untreated as the results text implies.

The way the authors have this section laid out, it is very difficult to keep track of what is statistically valid in terms of the comparisons they are making. They also seem to be making reference to individual transcripts that are "statistically significant" but only to an uncorrected p of 0.05. This is problematic; it is one thing to use an uncorrected p threshold for transcripts that go into a pathway analysis, where the purpose is just a broad overview of the pathways engaged. It is quite another to focus in on specific transcripts that are the result of a statistical thresholding for which the false discovery rate was not controlled for.

This section needs substantial revision to make the comparisons, resulting transcripts, and pathways extremely clear. The authors also need to deposit the differential expression output with the raw sequencing data – there is no way to check their data in its current form without me processing it from raw reads to differential expression myself.

Dear Editor,

On behalf of all coauthors I thank you and reviewers very much for taking care of and thoroughly reviewing the manuscript once again that helped us improve it. All the comments we received on this manuscript have been taken into account in improving the quality of the article, and we present our reply to each of them separately. All changes in manuscript are highlighted in red.

Below is our point-by-point detailed response to the reviewers' comments.

Reviewer #2 (Remarks to the Author):

For the most part the authors have addressed my previous concerns. However, their edits have raised new ones, including spots that were changed that were not marked.

In the paragraph describing the pathway analysis (“To determine what cellular processes are affected by selected...”), the initial submission described the threshold for statistically significant enrichment as pathways passing an FDR of less than 0.05. However, in this resubmitted manuscript, that has been changed to p less than 0.05, suggesting that they used an uncorrected p-value threshold; the figure legend matches this. Using a nominal p-value cutoff for pathway enrichment would not be appropriate. See Wijesooriya et al, PLoS Comp. Biol (2022). Furthermore, their expanded explanation of the statistical filtering performed for the transcriptome analysis also brings up more questions;

Response:

We corrected the Figure and highlighted the pathways passed the FDR threshold in red. Relevant corrections have also been included in the manuscript (Transcriptome analysis section).

- It is quite surprising that there were no transcripts passing an FDR of 0.05 in the 50-day old females but 745 passing that threshold in the 50-day old males. There seems to be a similar effect on survival from the combination treatment, and based on the survival graphs in figure 1, by 50 days of age the survival of treated vs wt flies seems to be similar in males and females, but the transcript expression patterns are so different that there are ~700 genes passing an FDR of 0.05 in the males but none in the females? This seems like it should be investigated further,

especially given that this treatment yields 388 transcripts passing an FDR of 0.05 at 5 days of age.

Response:

Indeed, when analyzing the data of 50-day old females, we failed to detect genes with a FDR < 0.05. However, the figures presented for comparisons (treated vs untreated) only reflect the number of genes with a change in expression by more than 2 times and LogCPM > 1 (changes that we considered relevant). Complete lists of DE genes have been added to Supplementary Table 17.

- The shifting comparisons sometimes get a little confusing – the authors describe the DEG landscape between treated and untreated flies of both genotypes for 50 day old flies and 5 day old flies. But there is no mention of the comparable comparisons for 100 and 150 day old flies. Instead, the authors have shifted to comparing the genotypes. It would be helpful to perhaps have a table with all of the other comparisons, the threshold used for determining differential expression, and the corresponding number of DE genes.

Response:

The untreated (control) flies did not survive to 100 and 150 days. For clarity, we have made corrections in the text (Transcriptome analysis section).

There also was a mistake in the Supplementary Tables numbering... We renumbered the tables. In the revised version the list of experimental groups for RNA-seq is presented in the Supplementary Table S16.

In supplement table S16, the listed criteria are cutoff by the page, but it seems CPM > 0.1 and $p < 0.05$; however, in the results text, they list LogCPM > 1, $p < 0.05$ for the female 50 day old flies; the methods section suggests $p < 0.01$ and $\text{abs}(\text{LogFC}) > 0.3$ – which one is the correct threshold? Also, the text states “All analyzed variants are presented in Supplementary Table S16.” However, Table S16 only has E(z)/w vs w/w for 100- and 150-day old males and females. It doesn't seem to include the w/w treated vs. untreated as the results text implies.

10

Response:

In that table, we showed only top DE genes with more than 8-fold changes ($|\text{LogFC}| > 3$) between treated w/w and E(z)/w flies at 100 and 150 days of age. There was no additional filtering by

CPM. Previously, we have already described similar changes in these genes for flies with the $E(z)$ mutation. In the new version, we have decided to remove this table since the Supplementary Table 17 (lists of DEGs) includes among others a complete list of DE genes between treated ($E(z)/w$ vs w/w) for 100- and 150-day old flies. $p < 0.01$ and $\text{abs}(\text{LogFC}) > 0.3$ thresholds were used for the enrichment analysis. We used $p < 0.01$ since few genes passed $\text{FDR} < 0.05$ and since $p < 0.01$ threshold stronger than 0.05. The $\text{abs}(\text{LogFC}) > 0.3$ was chosen to ensure that very small changes are not taken into account.

In the revised version All analyzed variants are really presented in Supplementary Table S16. See above comment.

The way the authors have this section laid out, it is very difficult to keep track of what is statistically valid in terms of the comparisons they are making. They also seem to be making reference to individual transcripts that are “statistically significant” but only to an uncorrected p of 0.05. This is problematic; it is one thing to use an uncorrected p threshold for transcripts that go into a pathway analysis, where the purpose is just a broad overview of the pathways engaged. It is quite another to focus in on specific transcripts that are the result of a statistical thresholding for which the false discovery rate was not controlled for.

This section needs substantial revision to make the comparisons, resulting transcripts, and pathways extremely clear. The authors also need to deposit the differential expression output with the raw sequencing data – there is no way to check their data in its current form without me processing it from raw reads to differential expression myself.

Response:

Thank you for all the comments. We've made corrections in the text, Figures, and Supplementary material (See above) taking into account the FDR-value threshold. At the same time, in the Figure 6, we decided to leave the pathways that passed only the p -value threshold in order to display more information. All changes are highlighted in red. We have also added DEGs lists to Supplementary Table 17 so that readers can check the data.